



# New plume comparison metrics for the inversion of passive gases emissions

Pierre J. Vanderbecken[1], Joffrey Dumont Le Brazidec[1], Alban Farchi[1], Marc Bocquet[1], Yelva Roustan[1], Élise Potier[2], and Grégoire Broquet[2]

[1]CEREA, École des Ponts and EDF R&D, Île-de-France, France
[2]Laboratoire des Sciences du Climat et de l'Environnement, LSCE-IPSL (CEA-CNRS-UVSQ), Université Paris-Saclay, 91191 Gif-sur-Yvette, France

**Correspondence:** Vanderbecken Pierre (pierre.vanderbecken@enpc.fr)

**Abstract.** In the next few years, numerous satellites with high-resolution instruments dedicated to the imaging of atmospheric gaseous compounds will be launched, to finely monitor emissions of greenhouse gases and pollutants. Processing the resulting images of plumes from cities and industrial plants to infer the emissions of these sources can be challenging. In particular traditional atmospheric inversion techniques, relying on objective comparisons to simulations with atmospheric chemistry

5    transport models may poorly fit the observed plume due to modelling errors rather than due to uncertainties in the emissions.

The present article discusses how these images can be properly compared to simulated concentrations to limit the weight of modelling errors due to the meteorology used to analyse the images. For such comparisons, the usual pixel-wise $\mathcal{L}_2$ norm may not be a good option, because it is subject to the double penalty issue inherent to its local definition. This issue is characterised by a mutation of any position shift into significant amplitude discrepancies. To circumvent this issue, we propose to either

10   provide an upstream correction of the position misfit between the observed and simulated plumes in the usual $\mathcal{L}_2$ norm or to use a non-local metric based on the optimal transport theory, such as the Wasserstein distance.

All the metrics are evaluated using first a catalogue of analytical plumes and then more realistic plumes simulated with a mesoscale Eulerian atmospheric transport model, with an emphasis on the sensitivity of the metrics to position mismatch and the concentration values within the plumes. As expected, the metrics with the upstream correction are found to be less sensitive

15   to position errors in both analytical and realistic conditions. Furthermore, in realistic cases, we evaluate the weight of changes in the norm and the direction of the four-dimensional wind fields in our metric values. This comparison highlights the link between differences in the synoptic-scale winds direction and position error. It is found that discrepancies between two plume images due to wind direction errors in the meteorological conditions are less penalised by our new metrics with the upstream correction than without, thus avoiding the double penalty issue.



# 1 Introduction

Near real-time monitoring of atmospheric gaseous compounds at the scale of power plants, cities, regions and countries would allow decision makers to track the effectiveness of emission reduction policies in the context of climate change mitigation (Horowitz, 2016) or other voluntary emission reduction efforts. Inventories of emitted atmospheric gaseous compound are diverse in scale (Janssens-Maenhout et al., 2019; Kuenen et al., 2014) and methodology. The elaboration of comprehensive inventories generally combines various approaches based on a complex mixture of measurement techniques, database elaboration and numerical modelling. Despite the use of quality assurance and control verifications (Calvo Buendia et al., 2019a, b) the emissions fluxes can bear large uncertainties, depending on the species, on the countries or on the spatial scale (Cai et al., 2019; Hergoualc'h et al., 2021; Meinshausen et al., 2009; Pison et al., 2018; Solazzo et al., 2021). Furthermore, the delay between the emissions and the release of the corresponding inventory could be important due to a large amount of data to gather and aggregate. Even when the inventories are known to be accurate, they currently do not fulfil the need for real-time monitoring of the emissions at a regional scale. By observing from space the plumes of gases downwind of large cities and industrial plants, and atmospheric signals at a few to several hundred km scales, the new generation of high-resolution spectro-imagery may help address this need (Veefkind et al., 2012; Broquet et al., 2018). For instance, the future CO2M mission will provide images of $CO_2$ concentrations at a resolution of almost two kilometres square, which will enable the observation of urban scale pollutant plumes (Brunner et al., 2019; Kuhlmann et al., 2019, 2020). These new images can be directly used through fast methods to quantify the mean emissions of sources (Varon et al., 2018, 2020; Hakkarainen et al., 2021). These fast methods require only the images to provide an estimation of the emissions, but, they do so, by assuming either simplified chemistry, transport or temporal variations of the emissions.

Here we focus on the use of such images to update the emissions sources on a smaller time scale. This can be done using an inverse method relying on comparisons between the images and the predictions of a chemical transport model (CTM). A better match between the observed concentration fields and the simulated one will be the result of a more accurate source. However, the CTM prediction is bounded by the meteorological conditions used. It takes as inputs temperature, pressure, winds, cloud cover fields, etc. Usually, these atmospheric fields are provided by predictions previously obtained with mesoscale numerical weather prediction models (Lian et al., 2018). The estimated atmospheric fields come with uncertainties, which in turn yield uncertainties in the simulated concentration fields. Within the inverse technique, the concentration fields derived from satellites and CTM models are usually compared pixel-wise. However, the relative weight of the meteorological uncertainties within the comparison between observation and simulation cannot be easily removed through pixel-wise comparison, and thus the emissions inventories that estimate our sources could be erroneously updated due to the approximated meteorology used in the simulations. This issue is shared in other fields (Dumont Le Brazidec et al., 2021; Farchi et al., 2016; Keil and Craig, 2007). Here, our view is that meteorology drives the position error between the plume observed and the plume simulated by the CTM. Thus our main goal is then to define a metric for the comparison that level down the position error to reduce the weight of meteorology uncertainties within the inversion.





To control the weight of position uncertainties in a comparison, several methods have been developed. They can be classified
according to their basic principles. A first category corresponds to methods that project the image on a specific basis using
either droplet or analogous decomposition to keep only the critical components of the field (Briggs and Levine, 1997). A
second method corresponds to the relaxation of the pixel-wise comparison and hence to the smoothing of discrepancies due to
the shifting between the fields (Ebert, 2008; Amodei et al., 2009). A third category is based on object, feature description of the
image, their identification and their comparison using image processing method (Davis et al., 2006; Lack et al., 2010; Davis,
2019). Such a method stresses the differences in terms of simple patterns and objects between the images and does not provide
a straightforward evaluation of the gradient with respect to the inputs of the model which is needed for inverse modelling. The
fourth category is based on an upstream distortion of the images to take into account the position error (Gilleland et al., 2009;
Gilleland, 2021; Hoffman et al., 1995; Hoffman and Grassotti, 1996). These approaches were designed to score and evaluate
the quality of a given weather forecast with better interpretability. We will follow this idea of a moving field and apply it to
our problem of plume comparison. Our options are either to correct upstream the simulated concentration field or to adopt an
optimal transport metric that weights the movement to superimpose the observed concentration field with the simulated one.

The objective of this paper is to develop a simple and efficient metric for urban-scale plume images which can level down
the difference due to the meteorology while fitting into an inverse framework (following Feyeux et al., 2018; Tamang et al.,
2022). Even though the methods could be used for other gaseous compounds, reactive atmospheric gaseous have a more
complex transport due to chemistry. For the sake of simplicity, we will consider the $CO_2$ since it is a passive tracer. Several
metric candidates are introduced and compared. From the baseline local $\mathcal{L}_2$ norm, a new metric with an upstream non-local
correction of position errors is described in section 2. In section 3, going further away from the local comparison, we use the
optimal transport theory to define the Wasserstein distance between two plumes and then to build a new metric freed from
position errors. The different metrics are then evaluated and compared on a database of analytical two-dimensional Gaussian
puff cases in section 4. The metrics are then compared on a realistic database of $CO_2$ plumes from a German power plant in
section 5. For both databases, the images and the simulations are computed using the same model, which allows us to monitor
the discrepancies seen by the metrics. In section 6 we describe the dependence of the four metrics on meteorology, before
concluding in section 7.

## 2 Local metrics and illustration of double penalty issue using analytical plumes

In this section, we start by introducing the notation in section 2.1 and then the Gaussian puff model used to simulate the plumes
in the analytical experiments in section 2.2. Furthermore, we assume that the plumes are already detected and separated from
the background noise and instrumental noise. These steps bring challenges that are outside the scope of this article. The $\mathcal{L}_2$
norm is then defined in section 2.3, with an emphasis on the double penalty issue. To deal with the double penalty issue
associated with the family of pixel-wise metrics such as the $\mathcal{L}_2$ norm, a second metric is proposed in section 2.4.



## 2.1 Discrete and continuous representation of an image

In the present article, we focus on two-dimensional images – typically of the total column of $CO_2$ concentration, or of ground level concentration field –, full (no mask due to filtered data or clouds), with a discretisation of $N$ pixels. An image can hence be represented by a vector $\boldsymbol{x} = (x_1, \ldots, x_N)^\top \in \mathbb{R}^N$.

It is also possible to obtain a continuous representation of the image using an interpolation (e.g. bilinear). In this case, the image is represented by a two-dimensional field $X : \mathbb{E} \to \mathbb{R}$ defined on a finite domain $\mathbb{E} \subset \mathbb{R}^2$. Without loss of generality, we can assume that $\mathbb{E} = [0, 1]^2$. Furthermore, the two-dimensional field $X$ can be extended to $\mathbb{R}^2$ by using zero padding outside the original domain $\mathbb{E}$. If needed, a smooth transition from $X$ to $0$ can be included to avoid a sharp gradient at the boundaries of the original domain $\mathbb{E}$.

For each metric definition, we will use either the discrete or the continuous representation of the images, but this will be explicitly mentioned.

## 2.2 Analytical plumes

Our Gaussian puff model is a simplified model of a concentration field (e.g. concentration at a given altitude or total column concentration in specific conditions). It has the advantage to yield analytical expressions for the Wasserstein metrics (see section 3). It is also a relevant case in transport modelling: the transport of a three-dimensional Gaussian puff is a simplified model to estimate the transport of non-reactive pollutants (Korsakissok and Mallet, 2009; Seigneur, 2019). A set of Gaussian puffs is used extensively in the following sections to illustrate and evaluate the metric candidates.

In the Gaussian puff model, we assume that $X$ is proportional to the probability density function (pdf) of the normal distribution $\mathcal{N}(\boldsymbol{\mu}, \boldsymbol{\Sigma})$:

$$X(\boldsymbol{x}) \propto \frac{1}{\sqrt{(2\pi)^2 |\boldsymbol{\Sigma}|}} \exp\left[ -\frac{1}{2}(\boldsymbol{x} - \boldsymbol{\mu})^\top \boldsymbol{\Sigma}^{-1}(\boldsymbol{x} - \boldsymbol{\mu}) \right], \tag{1}$$

where $\boldsymbol{\mu}$ and $\boldsymbol{\Sigma}$ are the mean and the covariance matrix , respectively. The operator $|.|$ is the determinant for square matrices. Also note that, since the covariance matrix $\boldsymbol{\Sigma}$ is positive definite, it can be factored as follows:

$$\boldsymbol{\Sigma} = \mathbf{R}(\theta) \boldsymbol{\Delta} \mathbf{R}(\theta)^\top, \tag{2}$$

where $\mathbf{R}(\theta)$ is the rotation matrix of angle $\theta$, the angle between the principal axis of the Gaussian and the $x$-axis, and where $\boldsymbol{\Delta}$ is a diagonal matrix with the variance along the two principal axes of the Gaussian. Two examples of puffs based on the Gaussian puff model are provided in Figure 1, panels (b) and (c).

## 2.3 The $\mathcal{L}_2$ norm and the double penalty issue

To compare two concentration fields, one can see to what extent the fields overlap. This provides a pixel-wise (i.e. local) assessment of the discrepancies. The $\mathcal{L}_2$ norm is then defined as the sum of the squared discrepancies. More specifically, the





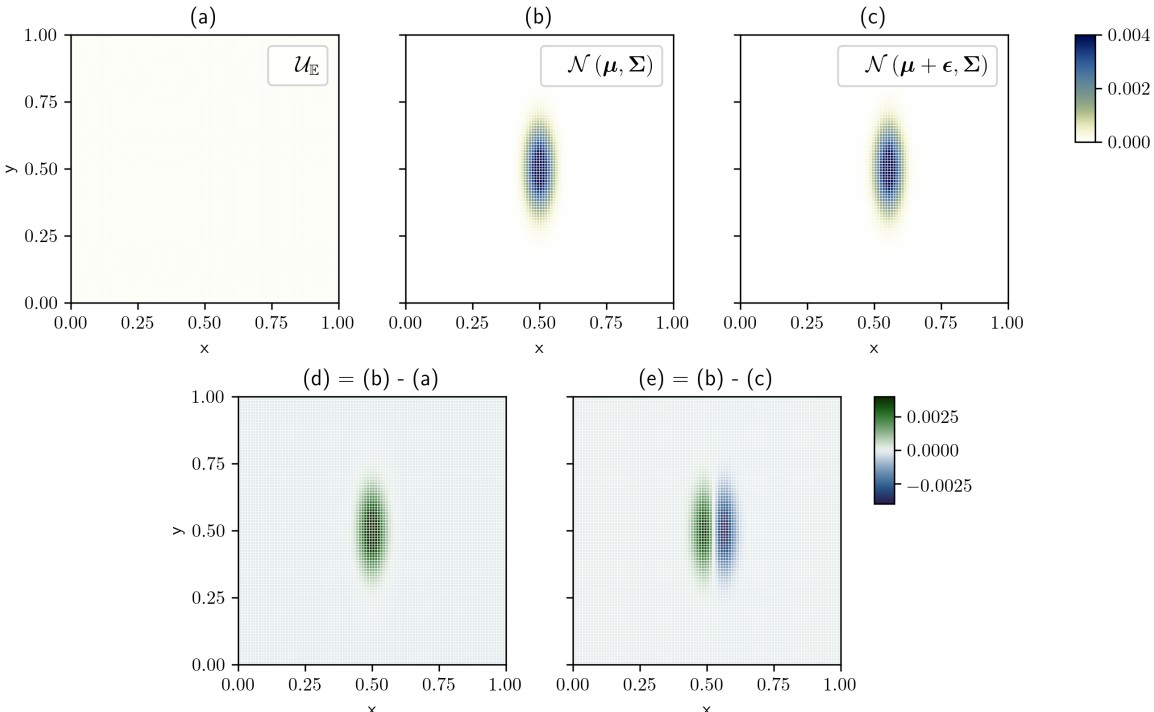

**Figure 1.** Example of pixel wise comparison. (a) Uniform concentration. (b) First Gaussian puff. (c) Second Gaussian puff, similar to (b) but shifted along the $x$ axis by $\epsilon = 0.054$. (d) Discrepancies between the concentration fields (b) and (a). (e) Discrepancies between the concentration fields (b) and (c).

$\mathcal{L}_2$ norm $d$ between two concentration fields $X_A$ and $X_B$ is defined as

$$d(X_A, X_B) \triangleq \sqrt{\frac{\int_{\mathbb{R}^2} \left[X_A(\boldsymbol{x}) - X_B(\boldsymbol{x})\right]^2 \mathrm{d}\boldsymbol{x}}{\int_{\mathbb{R}^2} \mathbf{1}_{\mathbb{E}} \mathrm{d}\boldsymbol{x}}}, \tag{3}$$

or

$$d(\boldsymbol{x}_A, \boldsymbol{x}_B) \triangleq \sqrt{\frac{1}{N} \sum_{n=1}^{N} (x_{A,n} - x_{B,n})^2}, \tag{4}$$

in the discrete case, where $\boldsymbol{x}_A$ and $\boldsymbol{x}_B$ are the two concentration vectors corresponding to the concentration fields $X_A$ and $X_B$. In the limit of an higher and higher resolution, the discrete formulation should converge towards the continuous formulation.

To identify the origin of the discrepancies, Feyeux et al. (2018) propose to split the difference between two fields into two categories: the position error and the amplitude error. A position error occurs when the two compared plumes are not located in the same place in the images. An amplitude error occurs when the two compared plumes are in the same place in the images but





**Table 1.** Comparison between the distances for the example in Figure 1. $d$ is the $\mathcal{L}_2$ norm, $d_F$ the $\mathcal{L}_2$ norm with upstream position correction as defined in section 2.4, $w$ the Wasserstein distance (section 3.1) and $w_F$ the Wasserstein distance with upstream position correction (section 3.4). The results are not provided with units since it depends on the metric used. The metrics $d$ and $d_F$ share the same units while $w$ and $w_F$ share an another one.

| Distance | (a) versus (b) | (b) versus (c) |
|---|---|---|
| $d$ | $48.80 \times 10^{-5}$ | $48.80 \times 10^{-5}$ |
| $d_F$ | $48.82 \times 10^{-5}$ | $11.59 \times 10^{-8}$ |
| $w$ | $32.68 \times 10^{-2}$ | $52.90 \times 10^{-3}$ |
| $w_F$ | $31.75 \times 10^{-2}$ | $93.13 \times 10^{-11}$ |

locally their pixels do not have the same values. With the $\mathcal{L}_2$ pixel-wise norm, all the discrepancies are seen as local amplitude errors. This property is illustrated in Figure 1, where a uniform concentration field $\mathcal{U}_{\mathbb{E}}$ is compared to two Gaussian puffs

shifted by $\epsilon = 0.054$ along the $x$ axis with respect to each other [1]. The values of the distance are reported in Table 1. In this case, a small position error is penalised by $d$ as much as an absence of plume: this is the so-called double penalty issue. The idea is that, instead of considering the cost of the translation, the metric adds the cost to set to zero all pixels from the first Gaussian puff to the cost to enhance the pixels at the translated location.

  In the following sections, we further extend the classification of Feyeux et al. (2018) by splitting the amplitude error into

two sub-categories: the scale error and the shape error. The scale error corresponds to the difference in total amplitude between two shape-matching fields. More practically, the difference between the sum of the compared image pixels. The shape error corresponds to the difference between the isocontours after removal of the scale error (i.e. normalisation) and position error (i.e. when both centres of mass and principal axes are superimposed) fields.

## 2.4 Local metric with non-local upstream position correction

We propose to address the double penalty issue while still relying on the $\mathcal{L}_2$ norm by applying an upstream correction of the position error to $d$. The position error can be seen as a combination of an orientation and a translation error. The orientation error corresponds to the differences that could be reduced by a rotation applied to two concentration fields sharing the same centre of mass location that maximises their overlapping. The translation error corresponds to the difference that could be reduced by a translation applied to two concentration fields.

The new distance is defined in a way that involves finding the rotation and translation that minimise $d$. The idea is that the rotation should cancel the orientation error and the translation should cancel the translation error. Let us consider the plane transformation $\mathbf{F}$ defined as follows:

$$\mathbf{F}(\boldsymbol{x}) = \boldsymbol{x_0} + \boldsymbol{x_t} + \mathbf{R}(\theta)[\boldsymbol{x} - \boldsymbol{x_0}], \tag{5}$$

---

[1]For this specific value of $\epsilon$, the $d$ distance between $\mathcal{U}_{\mathbb{E}}$ and the first plume is similar to the $d$ distance between the two plumes.





which corresponds to a translation of vector $\boldsymbol{x_t} = (x_t, y_t)^\top$, followed by a rotation of angle $\theta$ and of centre $\boldsymbol{x_0} + \boldsymbol{x_t}$, where $\boldsymbol{x_0} = (x_0, y_0)^\top$ is the position of the centre of mass of the plume before the transformation. The transformation $\mathbf{F}$ depends on three parameters: $(x_t, y_t, \theta)$. Note that this is an isometry of the plane. The optimal transformation should minimise

$$\mathcal{J}(x_t, y_t, \theta) \triangleq d^2(X_A, X_B \circ \mathbf{F}), \tag{6a}$$

$$= \int_{\mathbb{R}^2} [X_A(\boldsymbol{x}) - X_B(\mathbf{F}(\boldsymbol{x}))]^2 \, \mathrm{d}\boldsymbol{x} / \int_{\mathbb{R}^2} \mathbf{1}_\mathbb{E} \mathrm{d}\boldsymbol{x}. \tag{6b}$$

However, this cost function is constant for any transformation that moves all the mass of the $B$-plume outside the domain $\mathbb{E} = [0,1]^2$, where by construction $X_B$ is null. This would make the minimisation very difficult with gradient-based optimisation methods. For this reason, we add the following regularisation term to the cost function

$$\rho(x_t, y_t) \triangleq \begin{cases} (x_t^2 + y_t^2 - \frac{1}{2})^3 & \text{if } (x_t^2 + y_t^2) > 1/2, \\ 0 & \text{else,} \end{cases} \tag{7}$$

to penalise any transformation that moves the $B$-plume outside the domain $\mathbb{E}$. This regularisation does not affect the location of the minima of $d_F$. The final cost function is

$$\mathcal{J}(\theta, x_t, y_t) \triangleq \alpha \, d^2(X_A, X_B \circ \mathbf{F}) + \beta \, \rho(x_t, y_t) \tag{8}$$

where $\alpha$ is set to the average mass of the $A$- and $B$-plumes, and $\beta$ is set by trial and error to $10^4$. In practice, the cost function $\mathcal{J}$ can be minimised with the L-BFGS algorithm (Nocedal and Wright, 2006) that is based on the gradient of $\mathcal{J}$ with respect to all three parameters $\theta$, $x_t$, $y_t$, whose expression is given in appendix B. To compute the gradient, the spatial partial derivatives of the concentration field $X_B$ are needed. Hence, to ensure the continuity of the partial derivatives, we use a second-order bivariate spline interpolation to define the continuous concentration field $X_B$ from its original image $\mathbf{x}_b$. In order to avoid any issue due to the local non-convexity of the problem, we also provide a specific initialisation to the minimisation algorithm. The initial translation is then computed using the two centres of mass. Then we do orthogonal regressions to compute the principal axes of both $X_A$ and $X_B$. The initial $\theta$ is the angle between these axes.

Finally, with the optimal transformation $\mathbf{F}^*$, i.e. the one that minimises $\mathcal{J}$ defined by (8), the new distance, called $d_F$, is defined by

$$d_F(X_A, X_B) \triangleq d(X_A, X_B \circ \mathbf{F}^*). \tag{9}$$

For the example of Figure 1, the values of $d_F$ are reported in Table 1 and can be compared to the values of $d$. In the second case (distance between the two Gaussian puffs), $d_F$ is close to zero. The residual value is due to the finite resolution of the images. In the first case (distance between the Gaussian puff and the uniform concentration), $d_F$ stays similar to $d$ because any transformation $\mathbf{F}$ that keeps the plume in the domain is optimal.





## 3  Metrics based on optimal transport theory

In this section, we introduce the Wasserstein distance, the distance of the optimal transport, as a non-local alternative to the pixel-wise $\mathcal{L}_2$ norm.

### 3.1  Optimal transport and the Wasserstein distance

The optimal transport theory was first introduced in the XVII[th] century by Monge in his famous memoir (Monge, 1781). It is based on the idea that there exists a transport plan to move masses that minimises a given cost of transport. A wider view of the problem was proposed by Kantorovich (Kantorovich, 1942) using a probabilistic approach. The field has finally regained popularity in the last few decades, in particular with the generalisation by Villani (2009).

In this section, we follow the Kantorovich approach, which means that we will use the discrete representation (see sec-
tion 2.1). Moreover, the theory is defined only for vectors whose coefficients are non-negative and sum up to one. While the first condition is satisfied in our case (because we work with images of pollutant concentration), the second is not. Therefore, in the following instead of working with the concentration vectors $\boldsymbol{x}_A$ and $\boldsymbol{x}_B$, we will work with their normalised counterparts $\widehat{\boldsymbol{x}}_A$ and $\widehat{\boldsymbol{x}}_B$:

$$\widehat{\boldsymbol{x}} \triangleq \frac{\boldsymbol{x}}{\boldsymbol{x}^\top \mathbf{1}}, \tag{10}$$

where $\mathbf{1} \in \mathbb{R}^N$ is the vector full of ones and $\boldsymbol{x} \in \mathbb{R}^N$ is either $\boldsymbol{x}_A$ or $\boldsymbol{x}_B$.

The set of couplings $\mathbf{P}$ between $\widehat{\boldsymbol{x}}_A$ and $\widehat{\boldsymbol{x}}_B$ is defined by

$$\mathcal{U}(\widehat{\boldsymbol{x}}_A, \widehat{\boldsymbol{x}}_B) \triangleq \left\{ \mathbf{P} \in \mathbb{R}_+^{N \times N} \quad : \quad \mathbf{P}\mathbf{1} = \widehat{\boldsymbol{x}}_A \quad \text{and} \quad \mathbf{P}^\top \mathbf{1} = \widehat{\boldsymbol{x}}_B \right\}. \tag{11}$$

Note that $\mathcal{U}(\widehat{\boldsymbol{x}}_A, \widehat{\boldsymbol{x}}_B)$ is not empty because $\mathbf{P} = \widehat{\boldsymbol{x}}_A \widehat{\boldsymbol{x}}_B^\top$ is a coupling between $\widehat{\boldsymbol{x}}_A$ and $\widehat{\boldsymbol{x}}_B$. The cost of a coupling $\mathbf{P} \in \mathcal{U}(\widehat{\boldsymbol{x}}_A, \widehat{\boldsymbol{x}}_B)$ is defined by

$$\mathcal{J}(\mathbf{P}) = \sum_{i,j=1}^{N} C_{i,j} P_{i,j}, \tag{12}$$

where $C_{i,j} \geq 0$ is the $(i,j)$-element of the cost matrix $\mathbf{C}$ penalising the transport between $\widehat{\boldsymbol{x}}_A$ and $\widehat{\boldsymbol{x}}_B$. Here, it is chosen to be the square of the Euclidean distance between the $i$-th and $j$-th pixels of the original image. For this specific choice, the cost function $\mathcal{J}$ defined by eq. (12) has a minimum, which is obtained for a unique coupling $\mathbf{P}^*$. The Wasserstein distance, the distance of the optimal transport, is then defined by

$$w(\widehat{\boldsymbol{x}}_A, \widehat{\boldsymbol{x}}_B) = \sqrt{\sum_{i,j=1}^{N} C_{i,j} P_{i,j}^*} \tag{13}$$

and it is actually a distance according to the mathematical definition. The proofs of these statements can be found in optimal transport textbooks (e.g., Villani, 2009).





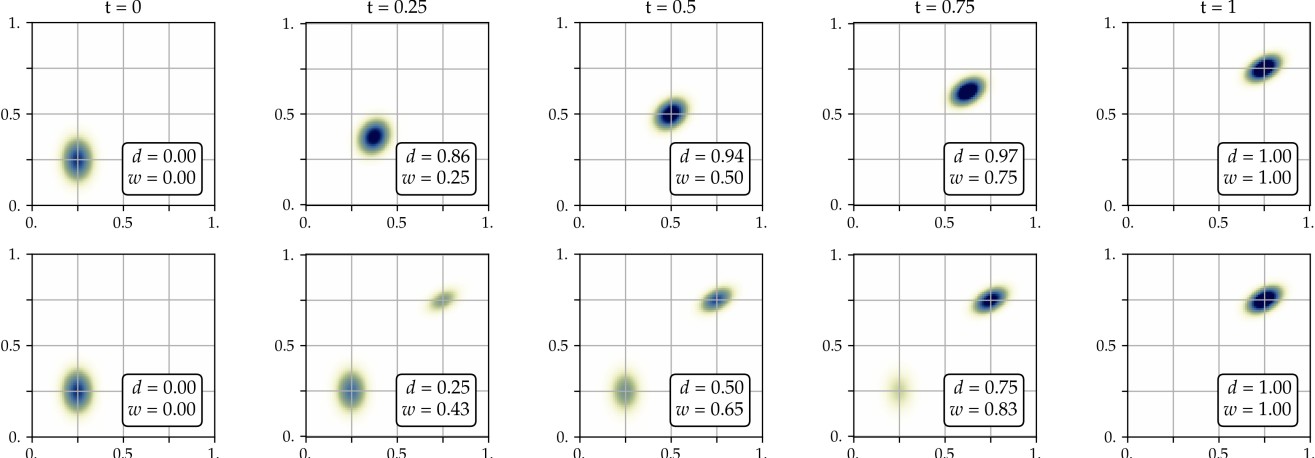

**Figure 2.** Comparison between the optimal transport interpolation (top panels) and the liner interpolation (bottom panels). In both cases, we interpolate between two puffs using a pseudo time ranging from $t = 0$ (interpolated puff equal to the first puff) to $t = 1$ (interpolated puff equal to the second puff). In each panel, the legend indicates both the $w$ and $d$ distances between the first puff and the interpolated puff, normalised by the distance between the first and second puff. By construction, for the optimal transport interpolation $w$ linearly grows with $t$, and for the linear interpolation $d$ linearly grows with $t$.

Two interesting properties of the Wasserstein distance can be highlighted. First, this metric is defined for normalised vectors only. This means in our case that the difference in total mass between two images is entirely ignored. Alternative solutions have been proposed to take into account this difference, e.g. the one proposed by Farchi et al. (2016) or the use of unbalanced optimal transport (Chizat et al., 2018), but this is beyond the scope of the present study.

Second, following Benamou and Brenier (2000), it is possible to define an optimal transport interpolation between $\widehat{\boldsymbol{x}}_A$ and $\widehat{\boldsymbol{x}}_B$. This optimal transport interpolation can help us visualise the idea of vicinity according to $w$. An example is shown in Figure 2 for two Gaussian puffs. In the case of the optimal transport interpolation, the $w$ distance between the first puff and the interpolated puff is linearly growing (by the construction of the interpolation), while the increase of the $d$ distance is at first very steep. In some sense, this behaviour was expected since the first puff and the interpolated puff are quickly separated from each other. In the case of the linear interpolation, the same phenomenon happens: the $d$ distance is linearly growing (by the construction of the interpolation), while the increase of the $w$ is steeper, but not as steep as the increase of $d$ in the first case. This shows that the Wasserstein distance $w$ is a softer metric than the distance $d$ and evaluates discrepancies more fairly especially those due to the position error.

## 3.2 Sinkhorn's algorithm

To compute the Wasserstein distance, we have to determine the optimal coupling matrix $\mathbf{P}^*$ by minimising $\mathcal{J}$ defined by eq. (12). The convexity of the cost function $\mathcal{J}$ is not guaranteed, thus it is usual (see, e.g., Peyré and Cuturi, 2019, and





references therein) to add the following entropic regularisation:

$$\mathcal{H}(\mathbf{P}) \triangleq - \sum_{i,j=1}^{N} P_{i,j} \left( \ln P_{i,j} - 1 \right). \tag{14}$$

The objective function to minimise becomes:

$$\mathcal{J}^{\epsilon}(\mathbf{P}) = \sum_{i,j=1}^{N} P_{i,j} C_{i,j} + \epsilon \sum_{i,j=1}^{N} P_{i,j} \left( \ln P_{i,j} - 1 \right), \tag{15}$$

under the same constraint $\mathbf{P} \in \mathcal{U}(\widehat{\boldsymbol{x}}_A, \widehat{\boldsymbol{x}}_B)$. The solution of the regularised problem is an approximation of the Wasserstein distance. When $\epsilon \to 0$ it converges toward the exact value of the Wasserstein distance $w(\widehat{\boldsymbol{x}}_A, \widehat{\boldsymbol{x}}_B)$ and when $\epsilon \to \infty$ the optimal coupling matrix converges toward $\mathbf{P} = \widehat{\boldsymbol{x}}_A \widehat{\boldsymbol{x}}_B^{\top}$.

It is possible to show that minimising eq. (15) is equivalent to minimising the Kullback–Leibler divergence between $\mathbf{P} \in \mathcal{U}(\widehat{\boldsymbol{x}}_A, \widehat{\boldsymbol{x}}_B)$ and the Gibbs kernel $\mathbf{K} = \exp(-\mathbf{C}/\epsilon)$, where the exponential is applied entry-wise, which is given by

$$\mathrm{KL}(\mathbf{P}|\mathbf{K}) = \sum_{i,j=1}^{N} P_{i,j} \ln \left( \frac{P_{i,j}}{K_{i,j}} \right) - P_{i,j} + K_{i,j}. \tag{16}$$

The advantage of this formulation it that this problem is known to admit a unique solution which is the projection of the Gibbs kernel $\mathbf{K}$ onto $\mathcal{U}(\widehat{\boldsymbol{x}}_A, \widehat{\boldsymbol{x}}_B)$. This unique solution can be written

$$\mathbf{P} = \boldsymbol{u}^{\top} \mathbf{K} \boldsymbol{v}, \tag{17}$$

where $\boldsymbol{u}$ and $\boldsymbol{v}$ are vectors with positive or null entries satisfying

$$\boldsymbol{u} \circ (\mathbf{K}\boldsymbol{v}) = \widehat{\boldsymbol{x}}_A, \tag{18a}$$

$$\boldsymbol{v} \circ (\mathbf{K}^{\top}\boldsymbol{u}) = \widehat{\boldsymbol{x}}_B. \tag{18b}$$

In these equations, $\circ$ is the Schur/Hadamard (i.e. entry-wise) product in $\mathbb{R}^N$.

The $(\boldsymbol{u}, \boldsymbol{v})$ factorisation is unique and can be easily found using the iterative update scheme proposed by Sinkhorn, where the $l$-th update is given by

$$\boldsymbol{u}^{(l+1)} = \frac{\widehat{\boldsymbol{x}}_A}{\mathbf{K}\boldsymbol{v}^{(l)}}, \tag{19a}$$

$$\boldsymbol{v}^{(l+1)} = \frac{\widehat{\boldsymbol{x}}_B}{\mathbf{K}^{\top}\boldsymbol{u}^{(l+1)}}, \tag{19b}$$

where $\div$ is the entry-wise division in $\mathbb{R}^N$.

### 3.3 Log-formulation of Sinkhorn's algorithm

Sinkhorn's algorithm provides a simple and quick solution to the optimal transport problem. However, this formulation raises technical issues. The first is that for small values of $\epsilon$ – which is what we are aiming for to be as close as possible to the true





optimal transport solution – the algorithm converges slowly[2]. To accelerate the convergence, we use a high value of $\epsilon$ and
progressively decrease it whenever $(\boldsymbol{u}, \boldsymbol{v})$ has converged. We will use this technique in our experiments.

Another numerical issue appears when $\epsilon$ is small compared to the entries of $\mathbf{C}$. In this case, $\boldsymbol{u}$, $\boldsymbol{v}$, and $\mathbf{K}$ explode and cannot be computed with finite numerical precision. To address this issue, we adopt the log-Sinkhorn algorithm proposed by Peyré and Cuturi (2019), which is presented in the following lines.

Let us introduce $\boldsymbol{f}$ and $\boldsymbol{g}$ which are related to $\boldsymbol{u}$ and $\boldsymbol{v}$ by

$$u_i = \exp\left(f_i/\epsilon\right), \tag{20a}$$

$$v_j = \exp\left(g_j/\epsilon\right). \tag{20b}$$

Instead of updating $(\boldsymbol{u}, \boldsymbol{v})$ with Sinkhorn iteration eq. (19), we update $(\boldsymbol{f}, \boldsymbol{g})$ using

$$f_i^{(l+1)} = -\epsilon \ln\left[\sum_{j=1}^{N} \exp\left\{\frac{f_i^{(l)} + g_j^{(l)} - C_{i,j}}{\epsilon}\right\}\right] + f_i^{(l)} + \epsilon \ln \widehat{x}_{A,i}, \tag{21a}$$

$$g_j^{(l+1)} = -\epsilon \ln\left[\sum_{i=1}^{N} \exp\left\{\frac{f_i^{(l+1)} + g_j^{(l)} - C_{i,j}}{\epsilon}\right\}\right] + g_j^{(l)} + \epsilon \ln \widehat{x}_{B,j}. \tag{21b}$$

Combining the log-Sinkhorn algorithm while decreasing $\epsilon$ is not straightforward, because there are a lot of numerical decisions to make: intermediate and final values of $\epsilon$, convergence criteria, etc. After several tests, we ended up with Algorithm 1, which we found to be a good trade-off between speed and accuracy. The value of $\epsilon$ is progressively decreased from 1 to $10^{-5}$: each time the convergence criterion is met, $\epsilon$ is reduced by a factor of 10. In our case, the convergence criterion is met when the relative difference between the former and the current value of the Wasserstein distance falls below $\zeta = 5 \times 10^{-4}$. In addition,
we set a maximum number of Sinkhorn iterations of 200 per value of $\epsilon$ to keep the computational cost under control. Finally, note that, for a given $\epsilon$, one can try to accelerate the convergence by using the averaging step proposed in Chizat et al. (2018) but this is beyond the scope of the present study.

### 3.4   Gaussian approximation and upstream correction

Following the derivation of section 2.4, we want to apply the same upstream correction of the position error to the Wasserstein
distance $w$. However, this would require the gradient of the Wasserstein distance $w$ with respect to each one of its inputs. The computation is not straightforward, even taking into account the log-Sinkhorn formulation developed in section 3.3. For this reason, we will use the Gaussian approximation, for which the Wasserstein distance has an analytical formula.

More specifically, we assume that we have two continuous concentration fields $X_A$ and $X_B$ that follow the Gaussian puff model:

$$X_A(\boldsymbol{x}) = \frac{1}{\sqrt{(2\pi)^2 |\boldsymbol{\Sigma}_A|}} \exp\left[-\frac{1}{2} (\boldsymbol{x} - \boldsymbol{\mu}_A)^\top \boldsymbol{\Sigma}_A^{-1} (\boldsymbol{x} - \boldsymbol{\mu}_A)\right], \tag{22a}$$

$$X_B(\boldsymbol{x}) = \frac{1}{\sqrt{(2\pi)^2 |\boldsymbol{\Sigma}_B|}} \exp\left[-\frac{1}{2} (\boldsymbol{x} - \boldsymbol{\mu}_B)^\top \boldsymbol{\Sigma}_B^{-1} (\boldsymbol{x} - \boldsymbol{\mu}_B)\right], \tag{22b}$$

---

[2]The convergence speed is measured here by the number of iterations.





---

**Algorithm 1** Log-Sinkhorn algorithm with decreasing $\epsilon$ to compute the Wasserstein distance.

---

**Parameters:** $\epsilon_0 = 1$, $\epsilon_* = 10^{-5}$, $\delta\epsilon = 10$, convergence criterion $\zeta = 5 \times 10^{-4}$, maximum number of iterations $k_{\max} = 200$

**Input:** Cost matrix $\mathbf{C}$, Normalised concentration vectors $\widehat{\boldsymbol{x}}_A$ and $\widehat{\boldsymbol{x}}_B$

1: $\boldsymbol{f} \leftarrow \boldsymbol{0}$

2: $\boldsymbol{g} \leftarrow \boldsymbol{0}$

3: $\epsilon \leftarrow \epsilon_0$      ▷ *Initialise $\epsilon$*

4: **while** $\epsilon \geq \epsilon_*$ **do**

5:     $k \leftarrow 0$      ▷ *Number of iterations*

6:     $w \leftarrow 10^5$      ▷ *Initialise $w$*

7:     **repeat**

8:       $w^- \leftarrow w$      ▷ *Previous value of $w$*

9:       **for** $i = 1$ **to** $N$ **do**

10:        $f_i \leftarrow -\epsilon \ln \left[ \sum_{j=1}^{N} \exp \left\{ \frac{f_i^{(l)} + g_j^{(l)} - C_{i,j}}{\epsilon} \right\} \right] + f_i^{(l)} + \epsilon \ln \widehat{x}_{A,i}$

11:       **end for**

12:       **for** $j = 1$ **to** $N$ **do**

13:        $g_j \leftarrow -\epsilon \ln \left[ \sum_{i=1}^{N} \exp \left\{ \frac{f_i^{(l+1)} + g_j^{(l)} - C_{i,j}}{\epsilon} \right\} \right] + g_j^{(l)} + \epsilon \ln \widehat{x}_{B,j}$

14:       **end for**

15:       $\mathbf{P} \leftarrow \exp \left\{ \frac{\boldsymbol{f}\mathbf{1}^\top + \mathbf{1}\boldsymbol{g}^\top - \mathbf{C}}{\epsilon} \right\}$

16:       $w \leftarrow \sqrt{\sum_{i,j=1}^{N} C_{i,j} P_{i,j}}$      ▷ *Current value of $w$*

17:       $k \leftarrow k + 1$

18:     **until** $|w^- - w|/w < \zeta$ or $k \geq k_{\max}$      ▷ *Convergence criterion*

19:     $\epsilon \leftarrow \epsilon/\delta\epsilon$      ▷ *Progressively decrease $\epsilon$*

20: **end while**

21: **return** Wasserstein distance $w$

---

with $\boldsymbol{\Sigma}_A$ and $\boldsymbol{\Sigma}_B$ given by

$$\boldsymbol{\Sigma}_A = \mathbf{R}(\theta_A)\boldsymbol{\Delta}_A\mathbf{R}(\theta_A)^\top, \tag{23a}$$

$$\boldsymbol{\Sigma}_B = \mathbf{R}(\theta_B)\boldsymbol{\Delta}_B\mathbf{R}(\theta_B)^\top. \tag{23b}$$

In this case, the squared Wasserstein distance between $X_A$ and $X_B$ is given by[3]

$$w^2(X_A, X_B) = \|\boldsymbol{\mu}_A - \boldsymbol{\mu}_B\|^2 + \mathrm{Tr}(\boldsymbol{\Sigma}_A + \boldsymbol{\Sigma}_B) - 2\,\mathrm{Tr}\left( \left[ \boldsymbol{\Sigma}_A^{1/2}\boldsymbol{\Sigma}_B\boldsymbol{\Sigma}_A^{1/2} \right]^{\frac{1}{2}} \right). \tag{24}$$

Following the approach of section 2.4, let us now apply the plane transformation $\mathbf{F}$ given by eq. (5) to $X_B$. The squared Wasserstein distance becomes

$$w^2(X_A, X_B \circ \mathbf{F}) = \|\boldsymbol{\mu}_A - \boldsymbol{\mu}_B + \boldsymbol{x}_t\|^2 + \mathrm{Tr}(\boldsymbol{\Delta}_A + \boldsymbol{\Delta}_B) - 2\,\mathrm{Tr}\left[ \boldsymbol{\Delta}_A^{1/2}\mathbf{R}(\theta + \theta_B - \theta_A)\boldsymbol{\Delta}_B\mathbf{R}^\top(\theta + \theta_B - \theta_A)\boldsymbol{\Delta}_A^{1/2} \right]^{\frac{1}{2}}, \tag{25}$$

---

[3]By construction, $X_A$ and $X_B$ are normalised, in such a way that we do not need to renormalise them to be able to compute the Wasserstein distance.





which depends on $x_t$, $y_t$, and $\theta$, the three parameters of $\mathbf{F}$. It can be shown (see appendix C) that $w^2(X_A, X_B \circ \mathbf{F})$ reaches its minimum when $\boldsymbol{x}_t = \boldsymbol{\mu}_A - \boldsymbol{\mu}_B$ and $\theta = \theta_A - \theta_B$ (modulo $\pi$), in which case the distance is given by

$$w(X_A, X_B \circ \mathbf{F}) = \sqrt{\mathrm{Tr}(\boldsymbol{\Delta}_A + \boldsymbol{\Delta}_B) - 2\,\mathrm{Tr}\left[(\boldsymbol{\Delta}_A \boldsymbol{\Delta}_B)^{\frac{1}{2}}\right]}, \tag{26a}$$

$$= \sqrt{\mathrm{Tr}\left[\left(\boldsymbol{\Delta}_A^{\frac{1}{2}} - \boldsymbol{\Delta}_B^{\frac{1}{2}}\right)^2\right]}, \tag{26b}$$

which is known as the Hellinger distance between $X_A$ and $X_B$ (Peyré and Cuturi, 2019). By construction, this distance
estimates the shape error between $X_A$ and $X_B$ since the translation, the rotation and the scale differences have been removed. In the following, it will be written $w_F$ to point out the similarity between the relationship $d/d_F$ on the one hand and $w/w_F$ on the other hand.

In the case where $X_A$ and $X_B$ are not exactly Gaussian, we can still use the Gaussian puff model as an approximation. In this case, $w_F$ provides an approximation of the shape error.

Finally, an issue with both $w$ and $w_F$ is that they are normalised fields and thus they ignore the scale error, i.e. the difference of total mass between the images. As a consequence, these metrics cannot be used as such in an inversion framework. One way to address this issue is to add to $w$ and $w_F$ a term to represent the scale error. Using the discrete formalism, this term could be

$$d_{\mathrm{mass}}^2(\boldsymbol{x}_A, \boldsymbol{x}_B) \propto \left[1 - 2\frac{\sum\limits_{n=1}^{N} x_{A,n} \sum\limits_{n=1}^{N} x_{B,n}}{\left(\sum\limits_{n=1}^{N} x_{A,n}\right)^2 + \left(\sum\limits_{n=1}^{N} x_{B,n}\right)^2}\right]^2, \tag{27}$$

which is convex. The remaining question would then be the relative contribution of $w$ (or $w_F$) and $d_{\mathrm{mass}}$ in the final distance,
which is related to the following question: which kind of error (position, mass, etc.) should be penalised more? This kind of question is beyond the scope of the present article, which is why we only use $w$ and $w_F$ as is in our numerical experiments.

## 4 Comparison of the metric on analytical test cases

In this section, we evaluate and compare the metrics with a database of images built using a set of Gaussian puffs. The database is introduced in section 4.1, the computation of the non-local metrics is validated in section 4.2, and the behaviour of the
metrics on this Gaussian puffs database are compared in section 4.3.

### 4.1 Gaussian puffs database and experimental setup

The database consists of $10^4$ pairs of images constructed using Gaussian puffs and then discretised on the domain $\mathbb{E} = [0,1]^2$ using a finite resolution of $32 \times 32$ pixels. Each puff is parametrised by its mean $\boldsymbol{\mu}$ (two scalars) and its covariance matrix $\boldsymbol{\Sigma} = \mathbf{R}(\theta)\boldsymbol{\Delta}\mathbf{R}(\theta)^\top$ (three scalars: $\theta$ and both diagonal entries of $\boldsymbol{\Delta}$), which are randomly drawn as follows:

1. both components of $\boldsymbol{\mu}$ are uniformly drawn in $[0.15, 0.85]$;





2. $\theta$ is uniformly drawn in $[-\pi, \pi]$;

3. $\sigma_1$ and then $\sigma_2$, the two non-zero components of $\boldsymbol{\Delta}$ are drawn from a half-normal distribution with a standard deviation of 0.33. If needed, $\sigma_1$ and $\sigma_2$ are then swapped to ensure $\sigma_1 \geq \sigma_2$.

Ideally, the domain $\mathbb{E}$ should cover a large majority of the mass of each puff. In practice, more than 99% of the mass of a
Gaussian puff is covered by the $6\sigma_1 \times 6\sigma_2$ rectangle centred on $\boldsymbol{\mu}$ and oriented along the principal axes. For this reason, we repeat step 3 of the random draw until this $6\sigma_1 \times 6\sigma_2$ rectangle is included in the domain $\mathbb{E}$. In addition, the puffs should not be too small, which is why in our case when $6\sigma_1$ and $6\sigma_2$ are both smaller than 9 pixels, it is rejected and entirely re-drawn.

The characteristics of the database are shown in Figure 3. As expected, the distribution of $\|\boldsymbol{\mu}\|$ is close to Gaussian, the distribution of $\theta$ is close to uniform, and the distribution of $\sigma_1$ and then $\sigma_2$ are close to log-normal.

### 4.2  Validation of the implemented Sinkhorn algorithm

For our Gaussian puffs database, there are four different ways to compute the Wasserstein distance:

1. use the analytical formula eq. (24) with the exact values of $\boldsymbol{\mu}_{A,B}$ and $\boldsymbol{\Sigma}_{A,B}$; this approach will be called $w_{\text{th}}$;

2. use the analytical formula eq. (24) but with $\boldsymbol{\mu}_{A,B}$ and $\boldsymbol{\Sigma}_{A,B}$ being the sample mean and covariance of the $32 \times 32$-pixel images; this approach is closer to what will be pratically done for real image plume, extracting information only from
the image, and it will be called $w_{\text{num}}$;

3. use the Network Simplex algorithm (Bonneel et al., 2011) to find the exact solution of the optimal transport problem using the images like $w_{\text{num}}$; this approach will be called $w_{\text{emd}}$;

4. apply the log-Sinkhorn iterations using algorithm 1 using the images like $w_{\text{num}}$; this approach will be called $w_{\epsilon}$.

We have applied all four methods and the differences are shown in Figure 4. Note that $w_{\text{emd}}$ has been computed using the POT
library (Flamary et al., 2021).

The fractional bias over all pairs is no more than 5% when we compare $w_{\text{th}}$ to the other three methods of computing the Wasserstein distance. By contrast, $w_{\text{emd}}$ and $w_{\text{num}}$ are very close to each other. We have checked that the latter phenomenon is reduced when the resolution is increased. Therefore, we conclude that the gap between $w_{\text{th}}$ on the one hand, and $w_{\text{num}}$, $w_{\text{emd}}$, and $w_{\epsilon}$ on the other hand is not due to the estimation of the Wasserstein distance but results from the discretisation of the
problem with the $32 \times 32$ resolution (sampling errors). Figure 4 also shows that $w_{\epsilon}$ matches well $w_{\text{emd}}$, which validates our log-Sinkhorn implementation.

### 4.3  Correlation to the different error categories

In this subsection, we compare the behaviour of the metrics with respect to three error categories: the translation error, the orientation error, and the shape error. Note that the behaviour with respect to the scale error cannot be compared since the $w$
and $w_F$ distances use normalised images. We used the Pearson correlation as our main indicator of the strength of the link

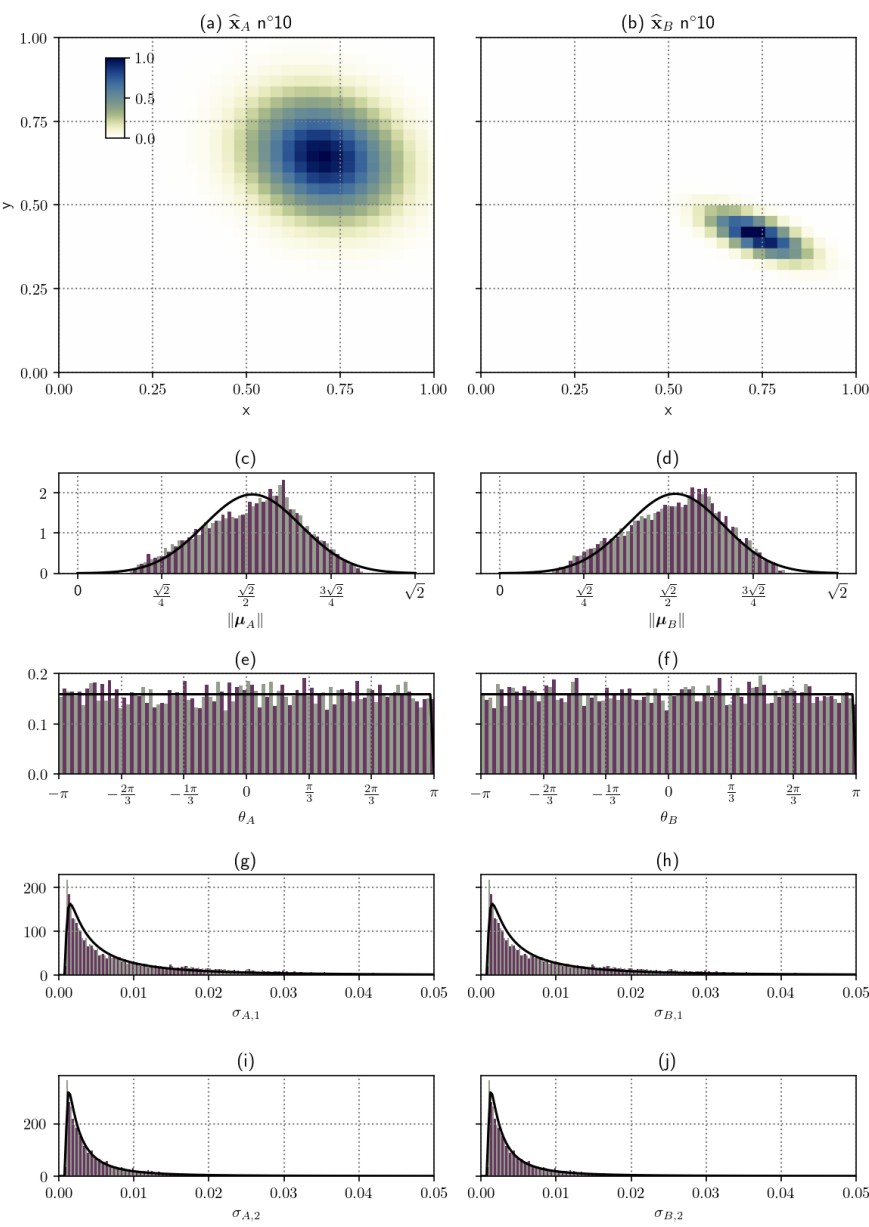

**Figure 3.** Characteristics of the Gaussian puffs database. (a, b) Images $A$ (left) and $B$ (right) number 10. (c-j) Histograms of $\|\boldsymbol{\mu}_A\|$ (c), $\|\boldsymbol{\mu}_B\|$ (d), $\theta_A$ (e), $\theta_B$ (f), $\sigma_{A,1}$ (g), $\sigma_{B,1}$ (h), $\sigma_{A,2}$ (i), and $\sigma_{B,2}$ (j).





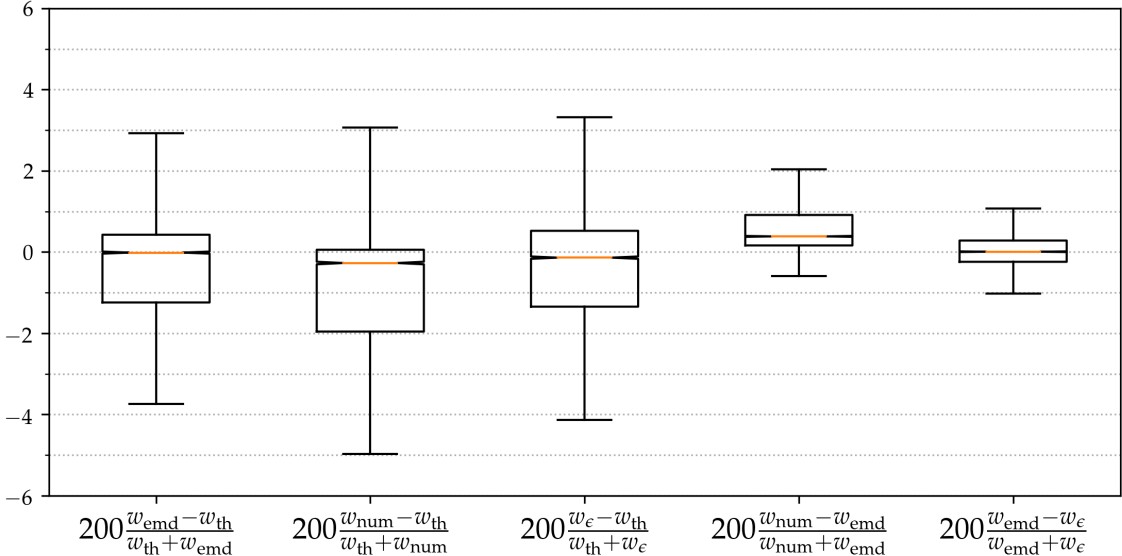

**Figure 4.** Comparison of the different ways to compute the Wasserstein distance over the Gaussian puffs database. Relative errors between $w_{\mathrm{emd}}$, $w_{\mathrm{th}}$, $w_{\mathrm{num}}$, and $w_\epsilon$ (c).

**Table 2.** Correlations between the distances $d$, $w$, $d_F$, and $w_F$ on the one hand, and the quantities $T$, $\theta$, and $H$ on the other hand for the $10^4$ cases in the Gaussian puffs database.

| | Pearson correlation | | |
|---|---|---|---|
| | $T$ | $\theta$ | $H$ |
| $d$ | 0.33 | 0.00 | $-0.11$ |
| $w$ | 0.97 | 0.00 | $-0.03$ |
| $d_F$ | 0.04 | $-0.01$ | 0.58 |
| $w_F$ | $-0.04$ | $-0.01$ | 0.65 |

between the behaviour of the metrics and the error category. The closer the norm of the Pearson correlation is to one, the more linearly the relation between the quantities is. If the Pearson correlation is positive, then the increase in an error category leads to an increase in the metric value, if negative it leads to a decrease, and if nearly null it means that the two quantities seem independent.

For each pair of images in the database, we define $T$ (for *translation*) as $\|\boldsymbol{\mu}_B - \boldsymbol{\mu}_A\|^2$. This quantity represents the translation error between both images. The correlation between $T$ and the four distances is reported in the first column of Table 2.

As expected, the Wasserstein distance $w$ is strongly correlated to $T$. The $\mathcal{L}_2$ norm $d$ is also showing a significant correlation of 0.33 to $T$, highlighting a likely dependency. Both $d_F$ and $w_F$ are designed to be released from the position error and, in



particular, the translation error. This property is confirmed by the very low correlation between $T$ on the one hand and $d_F$ and $w_F$ on the other hand. Additionally, the fact that $T$ is much more correlated to $d$ than to $d_F$ confirms that $d$ indeed depends on the $T$ quantity.

For each pair of images in the database, we define $\theta$ as $\|\theta_B - \theta_A\|$. This quantity represents the orientation error between both images. The correlation between $\theta$ and the four distances is reported in the second column of Table 2. The results shows also that there is no correlation between $\theta$ and any of the distances. In a sense, this shows that all the distances are, for our database, not sensitive to the orientation error.

For each pair of images in the database, we define $H$ as the Hellinger distance between $A$ and $B$, as given by eq. (26). This is actually very similar to $w_F$, but with one exception: $H$ uses the *theoretical* values of $\boldsymbol{\Delta}_A$ and $\boldsymbol{\Delta}_B$ (i.e. the ones that have been drawn) while $w_F$ uses the sample covariance of the $32 \times 32$-pixel images. This quantity represents the shape error between both images. The correlation between $H$ and the four distances is reported in the third column of Table 2.

Both $d$ and $w$ show a low correlation to $H$, which is not the case of $d_F$ and $w_F$. On the one hand, the correlation between $w_F$ and $H$ was highly expected from the definition of $H$. The remaining difference is due to the finite resolution of the images. On the other hand, the proportionality of $d_F$ with the $H$ was wanted but not assessed. By superimposing optimally the plumes, we removed the position error but $d_F$ remains sensitive to $H$, meaning we did not remove all errors. Thus such behaviour reflects our way of splitting the error. More generally, this comparison on the Gaussian puffs database confirms that both $d_F$ and $w_F$ are freed from the position error and seem to be driven by the shape error.

## 5 Comparison of the metric on realistic test cases

To go deeper in our analysis, we now compare the metrics using realistic plumes. This section follows the same organisation as section 4: we present the experimental setup in section 5.1, we validate the computation of the non-local metrics in section 5.2, and we compare the behaviour of the metrics on this specific database in section 5.3.

### 5.1 Simulation database and experimental setup

We use a simulation database of hourly 3D fields of $CO_2$ concentrations due to anthropogenic $CO_2$ emissions from the Neurath lignite-fired power plant (Germany, 51.04°N, 6.60°E). This database is composed of 14 days of 1h-emission pulses, from July 1 to 14, 2015, i.e. 336 plume transports. Plume transport occurs over fixed 24-hour windows. (from 00:00 to 24:00). Consequently, the later in the day the plumes are emitted the shorter they are tracked. The database is extracted from a larger one, over Western Europe, as described in Potier et al. (2022). Simulations were performed with the CHIMERE Eulerian transport model (Menut et al., 2013) driven by the Community Inversion Framework (CIF, Berchet et al., 2021). The horizontal grid resolution of the simulation domain (longitude: 6.82°W to 19.18°N; latitude: 42.0°N to 56.39°N, Fig. 5, Santaren et al., 2021) varies between 50 km and km. The Neurath power plant is located in the $2\,\text{km}\times2\,\text{km}$-resolution zoom (longitude: 1.25°W to 10.64°E; latitude: 47.45°N to 53.15°N). The vertical grid is composed of 29 pressure layers extending from the surface to 300 hPa (approximately 9 km above the ground level). CHIMERE is forced by meteorological variables at 9 km resolution





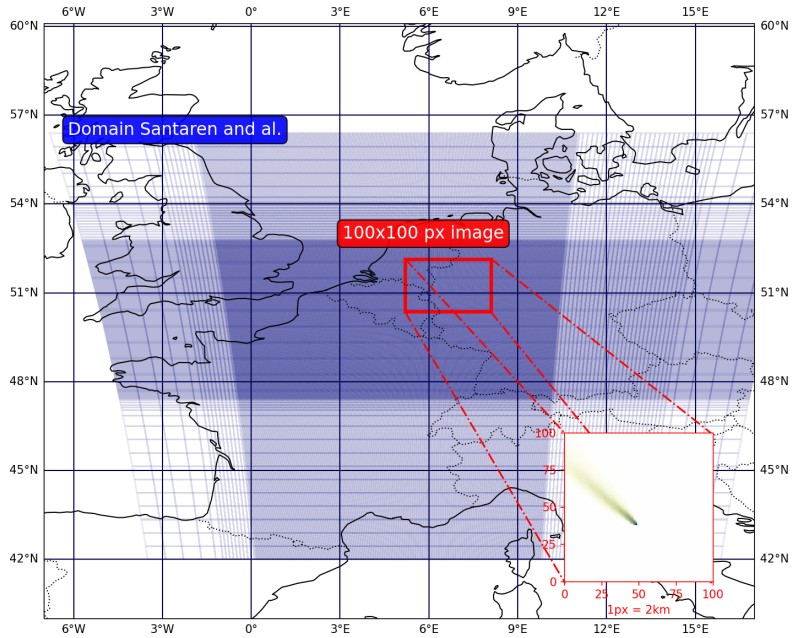

**Figure 5.** Experimental setup. Simulation domain in blue. Example of an $CO_2$ image used in red.

(Agusti-Panareda, 2018), provided by the European Centre for Medium-Range Weather Forecasts (ECMWF) for the $CO_2$ Human Emissions project (CHE, https://www.che-project.eu/). The $CO_2$ emissions from the Neurath power plant are extracted from the $\sim 1$ km ($1/60° \times 1/120°$) resolution inventory (TNO_GHGco_1x1km_v1_1) of the annual emissions produced by the Netherlands Organisation for Applied Scientific Research (TNO) over Europe for the year 2015 (Denier van der Gon et al.,

2017; Super et al., 2020). The temporal disaggregation at the hourly scale is based on coefficients provided with the TNO inventories for the sector A-Public Power, in the Gridded Nomenclature For Reporting (GNFR) of the United Nations Framework Convention on Climate Change (UNFCCC). Emissions are projected on the CHIMERE vertical grid with coefficients corresponding to this A GNFR sector (Bieser et al., 2011), also provided with the TNO inventories.

We ensure that the same daily profile is applied to the source emission, then for a given hour of the day, the difference

between two simulated plumes is the meteorological state. We build a database that regroups per pair two simulated plumes at a given hour but from different days (example: day 1 hour 10 versus day 3 hour 10). To get a realistic two-dimensional concentration field, we compute the vertical mean of the concentration weighted by the width of the vertical levels. We ignore the first two hours of the simulation, to ensure that a plume appears in the image. This leaves $2,093$ pairs of distinct plumes. The images are cropped to 100x100 pixels (here 1 pixel is equal to 2 km square cell of the simulation) images to reduce the

computer resource requirements.





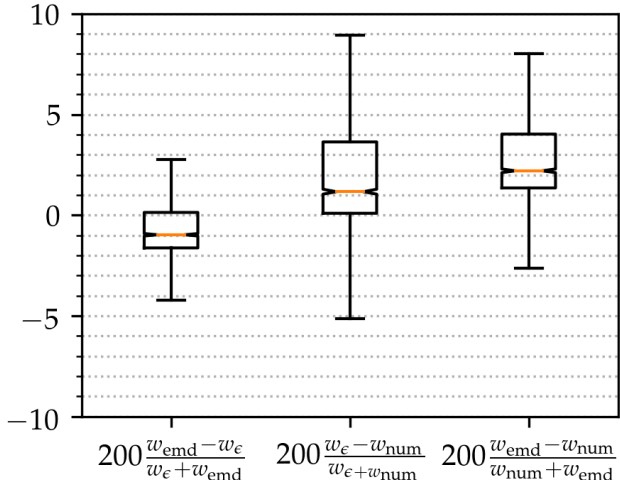

**Figure 6.** Comparison of the different ways to compute the Wasserstein distance over the realistic database. Relative errors between $w_{\mathrm{emd}}$, $w_{\mathrm{num}}$, and $w_\epsilon$ (c).

## 5.2 Comparison of the different estimations of the Wasserstein distance

We have applied all three methods and the differences are shown in Figure 6. The results show, as for the Gaussian puffs database, that $w_\epsilon$ and $w_{\mathrm{emd}}$ are close to each other, which once again validates our algorithm. Moreover, the results show that $w_{\mathrm{num}}$ is a reasonably good approximation of $w$ as well. The distance $w_{\mathrm{num}}$ makes the approximation that the images are Gaussian puffs, which is a strong approximation but allows for very quick computation. The values of $w_{\mathrm{num}}$ seem to be usually lower than those of $w_\epsilon$. This previous remark is in agreement with Theorem 2.1 from Gelbrich (1990). It is shown that, for elliptic contour distributions with given mean and covariance matrices, the distance between the two Gaussians with these respective parameters (i.e. $w_{\mathrm{num}}$) is a lower bound of the Wasserstein distance between the two distributions. We are not assured to work with plumes that are elliptic distributions. However, it seems to be a good direction to look at to explain and quantify, if possible, this negative bias. The understanding of the discrepancies between $w_{\mathrm{num}}$ and $w_\epsilon$ is needed to be able to substitute $w_{\mathrm{num}}$ to $w_\epsilon$, which is why we do not considered thereafter $w_{\mathrm{num}}$ in our comparisons.

## 5.3 Correlation to the different error categories

In this subsection, we compare the behaviour of the metrics with respect to the same three error categories than in section 4.3: the translation, orientation, and shape error. To do so, we keep the same quantities $T$, $\theta$, and $H$, with the notable exception that $H$ is now equal to $w_F$ because there is no *theoretical* covariance. The results are reported in Table 3.

While the correlation between $w$ and $T$ remains very strong, $d$ shows less correlation to $T$ than for the Gaussian puffs database. Both $d_F$ and $w_F$ are less correlated to $T$ than $d$ and $w$, respectively, but their correlation to $T$ is here higher than with





**Table 3.** Correlations between the distances $d$, $w$, $d_F$, and $w_F$ on the one hand, and the quantities $T$, $\theta$, and $H$ on the other hand for the $2,093$ cases in the realistic database.

| | Pearson correlation | | |
| --- | --- | --- | --- |
| | $T$ | $\theta$ | $H$ |
| $d$ | 0.19 | $-0.04$ | 0.32 |
| $w$ | 0.99 | 0.20 | 0.37 |
| $d_F$ | 0.12 | $-0.11$ | 0.41 |
| $w_F$ | 0.31 | 0.03 | 1.00 |

the Gaussian puffs database. Hence for this realistic database, both $d_F$ and $w_F$ are only partially freed from the translation error.

In this case, the correlations between the metrics and $\theta$ are higher than for the Gaussian puffs database but again do not prompt a clear conclusion.

    By construction, $w_F$ is equal to $H$, which yields a correlation of 1. Both $d$ and $w$ show a small correlation to $H$, which was not the case in the Gaussian puffs database. The correlation to $H$ is still higher for $d_F$, which was expected since $d_F$ is designed to be partially freed from the position error. This result, however, should be taken with caution because here, contrary

to the Gaussian puffs database, $H$ now only partially accounts for the shape error between two plumes.

    This second study with realistic cases shows that the behaviour of each metric slightly differs from what has been seen in the Gaussian case. Nevertheless, the results confirm that both $d_F$ and $w_F$ are partially freed from the position error while being still sensitive to the shape error, which is what we hoped for.

## 6   Sensitivity to the meteorological conditions

As stated in the introduction, the goal of this article is to develop and test metrics that can discriminate errors stemming from imperfect meteorology from other sources of discrepancies. Therefore, following the approach used in the previous sections, we define here four indicators that we consider representative of the difference in meteorological conditions between the two images. We then examine the correlation between these indicators, the previous indicators ($T$, $\theta$, and $H$), and the metrics in the case of the realistic database.

### 6.1   Definition of meteorological indicators

To simplify the analysis, we define four scalar indicators that characterise the meteorological conditions. These indicators focus on the direction and the norm of the wind as experienced by the pollutant during its transportation. For each image, we proceed as follows.





**Table 4.** Correlations between $\Delta E_N$, $\Delta E_D$, $\Delta S_N$, and $\Delta S_D$ on the one hand and $T$, $\theta$, and $H$ on the other hand for the $2,093$ cases in the realistic database.

| | Pearson correlation | | |
|---|---|---|---|
| | $T$ | $\theta$ | $H$ |
| $\Delta E_N$ | 0.39 | 0.23 | 0.15 |
| $\Delta E_D$ | 0.52 | $-0.02$ | 0.30 |
| $\Delta S_N$ | 0.09 | 0.04 | 0.09 |
| $\Delta S_D$ | 0.06 | 0.03 | 0.48 |

1. We first average the wind components (three-dimensional fields) in the vertical direction between the surface and the
planetary boundary layer (PBL) height. Indeed, the realistic database has been simulated with summer conditions and
     hence the plumes are assumed to be dispersed within the PBL. This results in two-dimensional fields for each time
     snapshot.

2. We compute the norm and the direction of the averaged winds. This results in two two-dimensional fields for each time
     snapshot.

3. We average the norm and the direction over the $100 \times 100$-pixel grid. This results in two scalars for each time snapshot.

4. We finally compute the time average and time standard deviation of the averaged norm and direction between midnight
     (the time at which the emissions started) and the time of the image. This results in four scalars: $E_N$ (averaged wind
     norm), $E_D$ (averaged wind direction), $S_N$ (deviation of wind norm) and $S_D$ (deviation of wind direction).

The meteorological indicators are then defined as the absolute differences in $E_N$, $E_D$, $S_N$, and $S_D$ between the two images
that are compared, simply written $\Delta E_N$, $\Delta E_D$, $\Delta S_N$, and $\Delta S_D$.

### 6.2 Correlation between the meteorological indicators and the error categories

Using the realistic database, we compute the correlation between $\Delta E_N$, $\Delta E_D$, $\Delta S_N$, and $\Delta S_D$ on the one hand and $T$, $\theta$, and
$H$ on the other hand. The idea is to see how differences in the meteorological conditions impact the position and amplitude
errors. The results are reported in Table 4.
One can notice that $T$ is mainly correlated to $\Delta E_D$ and a little less to $\Delta E_N$, while $\Delta S_D$ and $\Delta E_D$ are correlated to $H$.
This means that differences in meteorology like $\Delta E_D$ will likely induce both position error and shape error. Therefore, by
removing the position error, we only partially remove the meteorological impact on the differences. Explaining why $\Delta S_D$
induces differences in shape is straightforward, but explaining how $\Delta E_D$ induces differences in terms of translation instead of
orientation is not as so. A difference in the main direction of the plume (which translates into $\Delta E_D$) will move further away
the centres of mass from each other, and hence induce a larger $T$ (which is the distance between the two centres of mass). It
should be noted that a wind direction change that will keep superimposed the centre of mass will drive orientation error.





**Table 5.** Correlations between $\Delta E_N$, $\Delta E_D$, $\Delta S_N$, and $\Delta S_D$ on the one hand and the distances $d$, $w$, $d_F$, $w_F$, $d^*$ and $d_F^*$ on the other hand for the $2,093$ cases in the realistic database.

| | | | Pearson correlation | | | |
|---|---|---|---|---|---|---|
| | $d$ | $w$ | $d_F$ | $w_F$ | $d^*$ | $d_F^*$ |
| $\Delta E_N$ | $-0.09$ | $0.41$ | $-0.11$ | $0.15$ | $0.07$ | $0.02$ |
| $\Delta E_D$ | $0.14$ | $0.53$ | $0.14$ | $0.30$ | $0.24$ | $0.20$ |
| $\Delta S_N$ | $-0.03$ | $0.11$ | $0.00$ | $0.09$ | $-0.21$ | $-0.17$ |
| $\Delta S_D$ | $0.21$ | $0.09$ | $0.26$ | $0.48$ | $-0.03$ | $0.19$ |

### 6.3 Correlation between the meteorological indicators and the metrics

To conclude our study, we now compare the different metrics to the meteorological indicators. The results are reported in Table 5.

According to the correlations shown in Table 5, the metric $w$ is correlated to $\Delta E_D$ and $\Delta E_N$ indicators. It is expected since these meteorological changes tend to move the centre of mass and thus increase the translation error. The results show also that $w_F$ sees a drop in correlation to $\Delta E_D$ compared to $w$ while getting a correlation with respect to $\Delta S_D$. For optimal transport metrics, we can see that removing the position error does not always remove the sensitivity to a change in meteorology. It should be noticed that increasing in either $d$ or $d_F$ does not seem to be more correlated to our different meteorology indicators.

Such lack of correlation compared to the optimal transport theory metrics could result from the weight of the scale error in the distance definition. We normalised the plume the same way as we do for $w$ before computing the distance $d$ and $d_F$ leading us to the normalised image distances $d^*$ and $d_F^*$. First, $d^*$ and $d_F^*$ are more correlated than $d$ and $d_F$ to $\Delta E_D$ and $\Delta S_N$, showing that the scale error is masking the sensitivity of pixel-wise metrics with respect to meteorology. Second, $d_F^*$ gains significantly in correlation to $\Delta S_D$ compare to $d^*$, but remains as correlated to $\Delta E_D$ as $d^*$. Then the plane transformation

applied in $d_F^*$ allows a better alignment of the compared plumes, giving more weight to shape error induced by $\Delta S_D$, but does not compensate for all the changes resulting from $\Delta E_D$ or $\Delta E_N$.

The lack of correlation to our meteorological indicators for $d$ and $d_F$ seems appealing, but it is due to amplitude error held by a small number of highly concentrated pixels above the source for our cases (i.e. a hot spot). For similar cases, $d$ remains a good metric for updating the inventories. If the "hot spots" of the two images have amplitudes close to each other or there is no "hot-spot" but a large plume, $d$ and $d_F$ become more correlated to several meteorological changes making them less suitable.

Pixel-wise metrics seem to be better adapted to compare "hot-spot" and not highly extended plumes. A more versatile metric will be a weighted distance using the $w_F$, or at least a normalised $d_F^*$, which is not sensitive to all changes in meteorology, and a term that represents the scale error between the two images.





# 7 Conclusions

In this article, we discussed the use of new metrics for comparing passive gases plumes, practically $CO_2$ plumes, within an inverse framework aiming at the monitoring of pollutant emissions.

At first, we emphasised how critical the double penalty issue related to pixel-wise comparison is. The traditional $\mathcal{L}_2$ norm tends to overweight position errors mixing up with other sources of errors. In the context of source inversion, this results in an over-penalised comparison of concentration fields that are slightly shifted from each other, and the mixing makes it difficult to

evaluate the relative weight of different types of error afterwards. Yet, for us, the identification of the relative weight of the errors is critical since we want to level down the one due to meteorology and level up the one related to emissions. Assuming that most of the position error is driven by meteorology, we proposed to design metrics that are freed from position error. Following this goal, a pixel-wise metric with an upstream position correction was designed. This new metric has the advantage to keep the formalism of the $\mathcal{L}_2$ norm while being released from position errors. In addition, it is proposed to use a metric based on

the optimal transport theory: the Wasserstein distance. Focusing on the algorithmic aspects related to two-dimensional satellite images, we derived and validated a method to compute this metric. The Wasserstein metric is more sensitive to position errors but it is not hampered by the double penalty issue. To complete our catalogue of metrics, an optimal transport metric freed from position errors is proposed. It can be easily computed with a Gaussian approximation. This metric coincides with the Hellinger distance. Nevertheless, both optimal transport metrics rely on normalised images and thus are unaware of the difference in

total mass present in the plumes. The scale factor between the images is linearly related to emission fluxes which we want to estimate. This means that, within the inversion framework, the scale factor between the two images should be added and weighted independently.

These four metrics were compared on a specifically designed Gaussian puffs database and evaluated according to their correlations with respect to translation error, orientation error and shape error. The numerical experiments showed that the

resolution of the images tends to impact the optimal transport problem. As expected, the two metrics designed to be freed from position errors are not correlated to translation and orientation errors. The $\mathcal{L}_2$ norm and Wasserstein metrics are both correlated to the translation error. From this, we extended our tests to a realistic plume database. This second test series shows that, for a more complex plume geometry, the metrics are still correlated to the translation error. This implies that the new metrics are only partially freed from position errors.

Then we discussed the link between a position error and a variation within the mesoscale meteorology using the same realistic database. Designing relevant scalar indicators related to meteorological variance, we evaluated how specific changes in meteorological conditions lead to an increase in the distance between the plumes. We have seen that the meteorological changes can be correlated to position errors as well as amplitude errors between plumes. This means that removing the position error from the metrics will not make the comparison insensitive to a meteorological change. However, some metrics were

found to be more sensitive to specific changes in the meteorological conditions. For instance, while the Wasserstein metric is sensitive to changes in the main direction or intensity of the winds, the Hellinger metric is more sensitive to changes in the spread of the wind direction both in time and space. This provides guidelines to enlighten the choice of a metric for a given



meteorological situation. By composing with these new metrics freed from position error and additional scaling terms, we get more manageable metrics that will level down in the weight of modelling error due to the meteorology used in the comparison.

These metrics are used to quantify the proximity of a couple of plumes and could hence be used in an inverse framework, in particular for processing XCO2 images. The question of the impact of the meteorological changes on the metrics discussed here can be translated into another question: what importance do we give to each error category? We know that meteorological changes can result in position errors, and we strongly suspect that changes in the emission's temporal profile or vertical distribution can also yield position errors. In such a case, it would be interesting to evaluate the impact of the removal of the

position errors and if the amplitude errors carry enough information to compensate. We have seen that amplitude errors can also emerge from changes in meteorology. Thus further studies have to be undergone to evaluate the sensitivity of the metrics to either the emissions or the meteorology, to determine which error has to be more weighted from the perspective of monitoring the emissions. We have to make sure that by removing some sensitivity with respect to meteorology, we are not levelling down by the same factor the sensitivity with respect to the emissions.

For an operational purpose, optimising on non-local metrics is much more difficult than on pixel-wise metrics because it requires the computation of non-trivial gradients. The three non-local metrics that we proposed are parameterised. These parameters usually balance a trade-off between computational efficiency and accuracy. For the case of the pixel-wise distance with an upstream correction, this can have an impact on the optimum. Even though this study could be done with a personal computer, further computation optimisation developments are needed for operational use. Here we are only considering passive

tracer, but an extension of the study should be using these metrics for reactive pollutants. However, it requires quantifying the relative impact of chemistry on the shape, the scale and the position of the plume.

The key idea here is that meteorology is fixed and bounds our model predictions. We choose to develop metrics that aim to remove the weight of such bound within the comparison to the observation. We could instead consider that meteorology is not fixed and can be seen as additional degrees of freedom to estimate. Thus the Wasserstein metric is interesting due to its softer

behaviour than pixel-wise metrics but remains numerically costly. Yet, we have seen that approximating the plume by Gaussian puffs yields a cheap estimate of the true Wasserstein distance. To ease the computation, we suggest using an approximation of the Wasserstein distance, assuming Gaussian puff-like plumes or separable into a Gaussian mixture as in Chen et al. (2019); Delon and Desolneux (2020). But the relevance of these approximations has to be discussed when it comes to real, noisy, cloudy, plume images. This paper was a first step towards the use of smarter metrics to compare plume images to monitor

atmospheric gaseous compound emissions through an inverse method.

*Data availability.* All the data required to get the presented results are available on the zenodo deposit (Vanderbecken, 2022)



## Appendix A: Notation

| | Notations |
|---|---|
| $\boldsymbol{x}$ | Position vector in the image |
| $X_{A,B}$ | Continuous interpolation of the concentration field |
| $\boldsymbol{x}_{A,B}$ | Discrete representation of the concentration field |
| $\widehat{\boldsymbol{x}}_{A,B}$ | Normalised discrete concentration field |
| $\mathcal{N}(\boldsymbol{\mu}, \boldsymbol{\Sigma})$ | Normal distribution of mean $\boldsymbol{\mu}$ and error covariance matrix $\boldsymbol{\Sigma}$ |
| $\mathcal{U}_{\mathbb{E}}$ | Uniform distribution over the domain $\mathbb{E}$ |
| $\boldsymbol{\mu}$ | Always refers to a mean vector |
| $\boldsymbol{\Sigma}$ | Always refers to an error covariance matrix |
| $\boldsymbol{\Delta}$ | Diagonal matrix with the eigenvalues of $\boldsymbol{\Sigma}$ |
| $\mathbf{R}(\theta)$ | Rotation matrix of angle $\theta$ |
| $\boldsymbol{x}_t$ | Translation vector |
| $\boldsymbol{x}_0$ | Centre of mass coordinate vector |
| $\mathbf{F}$ | Transformation in the plane |
| $d$ | Usual pixel-wise Euclidean distance |
| $d_F$ | Pixel-wise distance with an upstream position correction |
| $w$ | Wasserstein distance |
| $w_F$ | Wasserstein distance with an upstream position correction |
| $w_{\text{emd}}$ | Earth mover distance |
| $w_{\epsilon}$ | Log-Sinkhorn approximation of the Wasserstein distance |
| $w_{\text{num}}$ | Wasserstein distance between two Gaussian puffs |
| $w_{\text{th}}$ | Analytical Wasserstein distance between two Gaussian puffs |
| $\epsilon$ | Weight of the entropic regularisation of the log-Sinkhorn algorithm |
| $\zeta$ | Convergence criterion for the log-Sinkhorn algorithm |
| $T$ | Translation length between the centre of mass of two plumes |
| $\theta$ | Rotation angle between the principal axes of two plumes |
| $H$ | Hellinger distance between the error covariance matrices of two plumes |
| $E_N$ | Mean wind speed seen by the plume averaged over the image domain and time |
| $E_D$ | Mean wind direction seen by the plume averaged over the image domain and time |
| $S_N$ | Standard deviation of the wind speed seen by the plume across the image domain and time |
| $S_D$ | Standard deviation of the wind direction seen by the plume across the image domain and time |





## Appendix B: Gradient of the cost function for $d_F$

To minimise eq. (8) we use the L-BFGS algorithm provided by the SciPy library. The algorithm explicitly uses the gradient of

the cost function $\mathcal{J}$ with respect to $\theta$, $x_t$, and $y_t$. The first term of this gradient – corresponding to $d^2\left(X_A, X_B \circ \mathbf{F}\right)$ – is given

by

$$\frac{\partial \mathcal{J}}{\partial \alpha} = -2 \int_{\mathbb{R}^2} \left[ X_A\left(\boldsymbol{x}\right) - X_A\left(\mathbf{F}\left(\boldsymbol{x}\right)\right) \right] \left[ \frac{\partial X_B}{\partial x}\left(\mathbf{F}\left(\boldsymbol{x}\right)\right) \cdot \frac{\partial F_x}{\partial \alpha} + \frac{\partial X_B}{\partial y}\left(\mathbf{F}\left(\boldsymbol{x}\right)\right) \cdot \frac{\partial F_y}{\partial \alpha} \right] \mathrm{d}\boldsymbol{x}, \tag{B1}$$

where $\alpha$ is either $\theta$, $x_t$, or $y_t$, $\boldsymbol{x} = (x,y)^\top$, and $\mathbf{F} = \left(F_x, F_y\right)^\top$. The partial derivatives of $X_B$ are given by the second image

(using the interpolation method), and the partial derivative of $F_x$ and $F_y$ are

$$\frac{\partial F_x}{\partial \theta} = -\left(x - x_0\right)\sin\theta - \left(y - y_0\right)\cos\theta, \qquad\qquad \frac{\partial F_y}{\partial \theta} = \left(x - x_0\right)\cos\theta - \left(y - y_0\right)\sin\theta, \tag{B2a}$$

$$\frac{\partial F_x}{\partial x_t} = 1, \qquad\qquad \frac{\partial F_y}{\partial x_t} = 0, \tag{B2b}$$

$$\frac{\partial F_x}{\partial y_t} = 0, \qquad\qquad \frac{\partial F_y}{\partial y_t} = 1. \tag{B2c}$$

## Appendix C: From the Wasserstein distance $w$ to the Hellinger distance $w_F$

Let us define the cost function

$$\mathcal{J}\left(x_t, y_t, \theta\right) \triangleq w^2\left(X_A, X_B \circ \mathbf{F}\right), \tag{C1}$$

where $w^2\left(X_A, X_B \circ \mathbf{F}\right)$ is given by eq. (25). The goal is to minimise $\mathcal{J}$. From eq. (25), we remark that $\mathcal{J}$ has three terms

$\mathcal{J} = \mathcal{J}_1 + \mathcal{J}_2 + \mathcal{J}_3$, with

$$\mathcal{J}_1 \triangleq \mathrm{Tr}(\boldsymbol{\Delta}_A + \boldsymbol{\Delta}_B), \tag{C2}$$

$$\mathcal{J}_2\left(x_t, y_t\right) \triangleq \left\| \boldsymbol{\mu}_A - \boldsymbol{\mu}_B + \boldsymbol{x}_t \right\|^2, \tag{C3}$$

$$\mathcal{J}_3\left(\theta\right) \triangleq -2\,\mathrm{Tr}\left[ \boldsymbol{\Delta}_A^{1/2}\mathbf{R}(\theta + \theta_B - \theta_A)\boldsymbol{\Delta}_B\mathbf{R}(\theta + \theta_B - \theta_A)^\top\boldsymbol{\Delta}_A^{1/2} \right]^{\frac{1}{2}}. \tag{C4}$$

Minimising $\mathcal{J}$ with respect to $\left(x_t, y_t, \theta\right)$ is equivalent to minimising $\mathcal{J}_2$ with respect to $\left(x_t, y_t\right)$ and minimising $\mathcal{J}_3$ with respect

to $\theta$. The minimum of $\mathcal{J}_2$ is 0 and is reached for $\boldsymbol{x}_t = \boldsymbol{\mu}_B - \boldsymbol{\mu}_A$. Let us now focus on the minimum of $\mathcal{J}_3$. For convenience,

we define

$$\mathbf{M}\left(\theta\right) \triangleq \boldsymbol{\Delta}_A^{1/2}\mathbf{R}(\theta + \theta_B - \theta_A)\boldsymbol{\Delta}_B\mathbf{R}(\theta + \theta_B - \theta_A)^\top\boldsymbol{\Delta}_A^{1/2}, \tag{C5}$$





in such a way that $\mathcal{J}_3(\theta) = -2\,\mathrm{Tr}\,\mathbf{M}(\theta)^{\frac{1}{2}}$. With our notation, we have

$$\mathbf{\Delta}_A = \begin{bmatrix} \sigma_{1,A} & 0 \\ 0 & \sigma_{2,A} \end{bmatrix}, \tag{C6a}$$

$$\mathbf{\Delta}_B = \begin{bmatrix} \sigma_{1,B} & 0 \\ 0 & \sigma_{2,B} \end{bmatrix}, \tag{C6b}$$

$$\mathbf{R}(\theta + \theta_B - \theta_A) = \begin{bmatrix} \cos\tilde{\theta} & -\sin\tilde{\theta} \\ \sin\tilde{\theta} & \cos\tilde{\theta} \end{bmatrix}, \tag{C6c}$$

where $\tilde{\theta} \triangleq \theta + \theta_B - \theta_A$, and hence

$$\mathbf{M}(\theta) = \begin{bmatrix} \sigma_{1,A}\sigma_{1,B}\cos^2\tilde{\theta} + \sigma_{1,A}\sigma_{2,B}\sin^2\tilde{\theta} & \sqrt{\sigma_{1,A}\sigma_{2,A}}\,(\sigma_{1,B} - \sigma_{2,B})\cos\tilde{\theta}\sin\tilde{\theta} \\ \sqrt{\sigma_{1,A}\sigma_{2,A}}\,(\sigma_{1,B} - \sigma_{2,B})\cos\tilde{\theta}\sin\tilde{\theta} & \sigma_{2,A}\sigma_{2,B}\cos^2\tilde{\theta} + \sigma_{2,A}\sigma_{1,B}\sin^2\tilde{\theta} \end{bmatrix}. \tag{C7}$$

By construction, $\mathbf{M}(\theta)$ is symmetric and positive definite, therefore it is diagonalisable with strictly positive eigenvalues $\lambda_\pm(\theta)$. As a consequence, we have

$$\mathrm{Tr}\,\mathbf{M}(\theta)^{\frac{1}{2}} = \sqrt{\lambda_+(\theta)} + \sqrt{\lambda_-(\theta)}. \tag{C8}$$

Let us now introduce the following ancillary quantities:

$$\alpha \triangleq \sigma_{1,A}\sigma_{1,B} + \sigma_{2,A}\sigma_{2,B}, \tag{C9a}$$

$$\beta \triangleq \sigma_{1,A}\sigma_{2,B} + \sigma_{2,A}\sigma_{1,B}, \tag{C9b}$$

$$\kappa(\theta) \triangleq \mathrm{Tr}\,\mathbf{M}(\theta) = \alpha\cos^2\tilde{\theta} + \beta\sin^2\tilde{\theta}, \tag{C9c}$$

$$\gamma(\theta) \triangleq \kappa^2(\theta) - 4\det\mathbf{M}(\theta) = \kappa^2(\theta) - 4\sigma_{1,A}\sigma_{1,B}\sigma_{2,A}\sigma_{2,B}. \tag{C9d}$$

Note that $\gamma(\theta)$ is the discriminant of the characteristic polynomial of $\mathbf{M}(\theta)$, which means that $\gamma(\theta) \geq 0$ because $\mathbf{M}(\theta)$ is
symmetric and positive definite. With these quantities, we have

$$\lambda_\pm(\theta) = \frac{1}{2}\left(\kappa(\theta) \pm \sqrt{\gamma(\theta)}\right). \tag{C10}$$

Let us first consider the case $\gamma(\theta) = 0$. In this case, $\lambda_+(\theta) = \lambda_-(\theta) \triangleq \lambda(\theta)$, in other words $\mathbf{M}(\theta) = \lambda(\theta)\mathbf{I}$. From the definition of $\mathbf{M}(\theta)$, eq. (C5), we deduce that

$$\mathbf{R}\left(\tilde{\theta}\right)\mathbf{\Delta}_B\mathbf{R}\left(\tilde{\theta}\right)^\top = \lambda(\theta)\mathbf{\Delta}_A, \tag{C11}$$

which enforces $\tilde{\theta} = 0$ (modulo $\pi$). This means that eq. (C7) simplifies into

$$\mathbf{M}(\theta) = \begin{bmatrix} \sigma_{1,A}\sigma_{1,B} & 0 \\ 0 & \sigma_{2,A}\sigma_{2,B} \end{bmatrix}, \tag{C12}$$




and hence $\lambda(\theta) = \sigma_{1,A}\sigma_{1,B} = \sigma_{2,A}\sigma_{2,B}$. Without loss of generality, we can assume in the definition of $\mathbf{\Delta_A}$ and $\theta_A$ that $0 < \sigma_{1,A} \le \sigma_{2,A}$ and the same for $B$[4]. This means that $\sigma_{1,A}\sigma_{1,B} = \sigma_{2,A}\sigma_{2,B}$ actually implies $\sigma_{1,A} = \sigma_{2,A}$ and $\sigma_{1,B} = \sigma_{2,B}$. In this case, the covariance matrices for $A$ and $B$ are isotropic and $\mathcal{J}_3$ does not actually depend on $\theta$.

Let us now consider the non-isotropic case: $0 < \sigma_{1,A} < \sigma_{2,A}$ and $0 < \sigma_{1,B} < \sigma_{2,B}$, which is the only case where $\mathcal{J}_3$ depends on $\theta$. In this case, we necessarily have $\gamma(\theta) > 0$. We can then take the derivative of $\mathcal{J}_3$ with respect to $\theta$:

$$-\frac{1}{2}\mathcal{J}_3'(\theta) = \frac{\lambda_+'(\theta)}{2\sqrt{\lambda_+(\theta)}} + \frac{\lambda_-'(\theta)}{2\sqrt{\lambda_-(\theta)}}, \tag{C13}$$

$$= \frac{\sqrt{\lambda_+(\theta)}\lambda_+'(\theta)}{2\lambda_+(\theta)} + \frac{\sqrt{\lambda_-(\theta)}\lambda_-'(\theta)}{2\lambda_-(\theta)}, \tag{C14}$$

$$= \frac{\sqrt{\lambda_+(\theta)}\left(\kappa'(\theta) + \frac{\gamma'(\theta)}{2\sqrt{\gamma(\theta)}}\right)}{4\lambda_+(\theta)} + \frac{\sqrt{\lambda_-(\theta)}\left(\kappa'(\theta) - \frac{\gamma'(\theta)}{2\sqrt{\gamma(\theta)}}\right)}{4\lambda_-(\theta)}, \tag{C15}$$

$$= \frac{\frac{\sqrt{\lambda_+(\theta)}}{\sqrt{\gamma(\theta)}}\left(\kappa'(\theta)\sqrt{\gamma(\theta)} + \frac{1}{2}\gamma'(\theta)\right)}{4\lambda_+(\theta)} + \frac{\frac{\sqrt{\lambda_-(\theta)}}{\sqrt{\gamma(\theta)}}\left(\kappa'(\theta)\sqrt{\gamma(\theta)} - \frac{1}{2}\gamma'(\theta)\right)}{4\lambda_-(\theta)}, \tag{C16}$$

$$= \frac{\frac{\sqrt{\lambda_+(\theta)}}{\sqrt{\gamma(\theta)}}\left(\kappa'(\theta)\sqrt{\gamma(\theta)} + \kappa'(\theta)\kappa(\theta)\right)}{4\lambda_+(\theta)} + \frac{\frac{\sqrt{\lambda_-(\theta)}}{\sqrt{\gamma(\theta)}}\left(\kappa'(\theta)\sqrt{\gamma(\theta)} - \kappa'(\theta)\kappa(\theta)\right)}{4\lambda_-(\theta)}, \tag{C17}$$

$$= \frac{2\frac{\sqrt{\lambda_+(\theta)}}{\sqrt{\gamma(\theta)}}\kappa'(\theta)\lambda_+(\theta)}{4\lambda_+(\theta)} - \frac{2\frac{\sqrt{\lambda_-(\theta)}}{\sqrt{\gamma(\theta)}}\kappa'(\theta)\lambda_-(\theta)}{4\lambda_-(\theta)}, \tag{C18}$$

$$= \kappa'(\theta) \cdot \frac{\sqrt{\lambda_+(\theta)} - \sqrt{\lambda_-(\theta)}}{2\sqrt{\gamma(\theta)}}, \tag{C19}$$

$$= (\beta - \alpha)\cos\tilde{\theta}\sin\tilde{\theta} \cdot \frac{\sqrt{\lambda_+(\theta)} - \sqrt{\lambda_-(\theta)}}{\sqrt{\gamma(\theta)}}, \tag{C20}$$

which is the product of three terms: $\beta - \alpha$, $\cos\tilde{\theta}\sin\tilde{\theta}$, and $\left(\sqrt{\lambda_+(\theta)} - \sqrt{\lambda_-(\theta)}\right)/\sqrt{\gamma(\theta)}$. The third term is always strictly positive because $\gamma(\theta) > 0$. The first term is always strictly negative because we have assumed that $\sigma_{1,A} < \sigma_{2,A}$ and $\sigma_{1,B} < \sigma_{2,B}$. Hence we only need to consider the second term $\cos\tilde{\theta}\sin\tilde{\theta}$ to conclude that the minima of $\mathcal{J}_3$ are met for $\tilde{\theta} = 0$ (modulo $\pi$) or equivalently $\theta = \theta_A - \theta_B$ (modulo $\pi$). In this case, $\mathbf{M}(\theta) = \mathbf{\Delta_A}\mathbf{\Delta_B}$ and $\mathcal{J}_3(\theta) = -2\,\mathrm{Tr}(\mathbf{\Delta_A}\mathbf{\Delta_B})$, which yields the correct formula for $w_F$, eq. (26). Finally note that this formula is also valid in the case where at least one of $A$ or $B$ is isotropic.

*Author contributions.* 1) writing process: mainly PV and EP with inputs from all co-authors, 2) System and experiment design: PV, JD, AF, MB, YR 3) Implementation: PV, 4) support in development and use of data: PV, EP 5) Analysis: mainly PV, JD, AF, MB with feedbacks from all co authors

---

[4]If this is not the case, we just have to change $\theta_A$ into $\theta_A + \pi$ to swap $\sigma_{1,A}$ and $\sigma_{2,A}$.



*Competing interests.* The authors declare that they have no conflict of interest.

*Acknowledgements.* This study has been funded by the national research project ANR-ARGONAUT N° ANR-19-CE01-0007 (Pollutants

600   and greenhouse gases emissions monitoring from space at high resolution). Joffrey Dumont le Brazidec is supported by the European Union's Horizon 2020 research and innovation programme under grant agreement N° 958927 (Prototype system for a Copernicus CO2service). All the figures are drawn using CVD-friendly colormaps. It was made possible using a Python wrapper around Fabio Crameri's perceptually uniform colormaps (Crameri, 2021), available here https://www.fabiocrameri.ch/colourmaps/. CEREA is a member of Institut Pierre-Simon Laplace (IPSL).





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
