# Peer review of "Simulated plumes freed from meteorological biases using smarter metrics?"

_Atmospheric Measurement Techniques, 2022_

## Author Comment (AC1)

**Simulated plumes freed from meteorology biases using smarter metrics? – response to the reviewers**

P. Vanderbecken, J. Dumont Le Brazidec, A. Farchi, M. Bocquet,
Y. Roustan, E. Potier, and G. Broquet

December 2022

**1 Answer to referee #2**

**1.** *The paper proposes a new measure for forecast performance that accounts for displacement methods. Overall, the methodology seems sound, but per comment 1 below, it is unclear if this approach is really new, or what the new contribution is. Moreover, it is unclear if the added complexity of the approach over the displacement methods (e.g., as discussed in doi: 10.1175/MWR-D-19-0256.1) adds enough value to warrant its use. Therefore, I recomment acceptance after the authors consider the comments below.*
→ First, we would like to thank you for your review and the relevance of your remarks and suggestions. This paper presented new metrics to compare pollutant plumes. Even though their description takes a significant part of this paper, the main goal is to see how the proposed metrics handle changes in meteorological conditions. To make this point clearer from the start, we propose a new title. We share your concerns about the benefit of using those metrics against the usual ones. Yet, for us, only the inversion results should be the judge of it. This will be discussed in a future paper.

**2.** *The method proposed is very similar to several of the field deformation approaches described by the cited Gilleland et al. paper and several since that time: e.g., see doi: 10.1016/S0022-1694(00)00343-7, doi: 10.1175/2010WAF2222365.1, doi: 10.5065/D62805JJ, doi: 10.1002/2012GL053964, and doi: 10.1175/2010WAF2222351.1 to name just a few. In particular, doi: 10.3402/tellusb.v68.31682 uses the Wasserstein distance. A thorough literature review and comparison of the differences and added utility of the present approach is necessary to put this work into the greater context of these deformation methods. As it is, it is not clear what the new contribution is over these other works.*
→ First of all, the different metrics proposed in this paper aim at handling position errors in a better way. This is indeed a similar goal to the field deformation approach or warping technique. But, as far as we know, the rotation that we consider as an orientation error,

which was suggested to us by practitioners, is usually not included in the position error and remains included within the shape error. The plane transform used in both $d_F$ and $w_F$ will conserve the shape of the plume which is not the case with the warping function used in the literature.

It is true that the Wasserstein metric has already been proposed by Farchi et al. (2016) to perform plume comparisons. The novelties here are that (i) we propose additional metrics beyond the Wasserstein distance, (ii) we provide a more systematic evaluation of these metrics, and (iii) we use a different algorithm to compute the Wasserstein distance (namely, the Sinkhorn algorithm).

In the revised manuscript, we have reformulated the introduction to explain these elements (L. 56 to 67), with hopefully more pertinent references to the literature.

**3.** *How does this approach address the issues outlined in doi: 10.5065/4px3-5a05 ?*
$\rightarrow$ In (doi: 10.5065/4px3-5a05), the authors share the analytical issues proposed by Davis, C et al. in (doi: 10.1175/2009WAF2222241.1). They correspond to pathological situations that occur when comparing features in images. We thank you for pointing out this relevant reference since any of our 10,000 analytical cases can be seen as a combination of these pathological situations. We add it in the manuscript (L. 309). Our metrics address these issues as described in the results of 10,000 analytic cases (section 4). To be more specific, both translation or rotation displacement are solved by $d_F$ and $w_F$ whether the plumes were initially overlapping or not, which is what we were aiming for in our case.

**4.** *The authors make reference to the measure's being fairer, but it is unclear what they mean by fair in the general concept of a fair verification measure.*
$\rightarrow$ Indeed, the word 'fairer' is subjective and should be avoided. We meant here that the translation error is linearly penalised by the Wasserstein distance while the L2-metric will reach a maximum when the plumes do not overlap. This has been explained in the revised manuscript (L. 211). Thank you for pointing this out.

---

## Author Comment (AC2)

**Simulated plumes freed from meteorology biases using smarter metrics? – response to the reviewers**

P. Vanderbecken, J. Dumont Le Brazidec, A. Farchi, M. Bocquet, Y. Roustan, E. Potier, and G. Broquet

December 2022

**1 Answer to referee #1**

**1.** *The authors discuss how to compare satellite observations to simulated concentrations by limiting the weight of modelling errors due to the meteorology used to analyze the observations. The manuscript presents a lot of equations to describe the math behind the method. I'm not sure I fully understand all the details, particularly section 3. But the work generally looks sound to me. I recommend the following revision.*
→ Thank you for your review and for your comments and suggestions on the manuscript. Your main concerns were on the clarity and the transparency of its assumptions, the method used and its goals.

**2.** *Our view is that meteorology drives the position error between the plume observed and the plume simulated by the CTM. This is the motivation of the work. However, I don't see how solid it is. There are several contributors to the errors. I don't see the reason why meteorology is the driver.*
→ There are likely several other contributors to the position errors. But if one deals with passive tracers and is under the assumption that the temporal variability of the emissions is known, then the position error will be driven by the transport and thus the meteorology. As we mention in the conclusion, it should be interesting to see how sensitive the plume is to the temporal profile in future work. We have modified the introduction to make this point clearer (L. 50-55).

**3.** *I recommend the author to add a flow chart to demonstrate the method.*
→ We have added a flow chart illustrating the different sources of errors when comparing two plumes (figure 2). Thank you for this nice suggestion.

**4.** *Line 7. It shall be analyze instead of analyse.*

→ Please note that the article is written in British English, for which the correct form should be 'analyse'.

**5.** *The concept of pixel-wise norm has been proposed without giving any introduction. Same as double penalty issue, upstream correction, non-local metric optimal transport theory. It will be difficult for readers without strong background for this very specific field to follow. I understand that it is difficult to give the definitions for all those items in a short abstract. I would like to encourage the authors to reconsider the necessity of keeping all those items and the possibility of rephrasing the paragraph in a more reader-friendly way.*

→ Following your suggestions, we have modified the abstract. We hope that it is now more reader-friendly.

**6.** *Line 40-45. Meteorology is not the only contributor to modelled bias. Such information seems missing from the text.*

→ Once again, you are right, this point requires some clarifications. Accordingly to your early suggestion 2, we have modified (L. 50-55). Indeed, there are others contributors to modelled bias, like the modelling error. But we assume our transport model perfect in the first place. This assumption can be relaxed in future studies.

**7.** *Line 51. What is "position error"?*

→ Thank you for spotting this issue: indeed, the term 'position error' was not defined at this point. We have reformulated the abstract, and 'position error' has been described as error due to a displacement between two identical plumes(L. 10 and 12), which we believe clearly refers to discrepancies between images due to a translation or rotation. Note that, in addition, the new flow chart (figure 2) should help clarify this term.

**8.** *Line 47. "the relative weight of the meteorological uncertainties within the comparison between observation and simulation cannot be easily removed through pixel-wise comparison". I don't quite understand the meaning of this sentence. It sounds like the aim of the comparison performed at pixel level is to remove meteorological uncertainties. Please try to rephrase it. Same for "This issue is shared in other fields". I'm not sure the sentence is clear to readers.*

→ The introduction has been reformulated, we hope that it is clearer now.

**9.** *Line 56. What is droplet or analogous decomposition? Please try to define before use.*

→ The idea of analogous decomposition is that for any state, one can look up in a given large database the typical recorded states close to this state according to some metric. This state can then be approximated as a combination of these closely related states. The droplets are function bases where any signal can be decomposed on.

However, note that in the end, we have removed the reference to these methods, as it introduces unnecessary complexity in the text.

**10.** *Line 58. What does "fileds" represent here? Line 64. What is a moving field?*
→ This part of the manuscript has been removed for the sake of simplicity and due to the previous comment.

---

## Author Response (AR3)

**Simulated plumes freed from meteorological biases using smarter metrics? – response to Lok Lamsal, associate editor**

P. Vanderbecken, J. Dumont Le Brazidec, A. Farchi, M. Bocquet, Y. Roustan, E. Potier, and G. Broquet

February 2023

**1 Answers to editor**

**1.** *Dear Authors, Thank you for submitting your revision and responses to reviewer's comments. I have now read, with great interest, the original manuscript, reviewer's comments, and revised version of the manuscript. I find that the work is interesting and it should be of value to the AMT readership. Based on my own reading, I feel that the manuscript needs some work mostly clarifying some statements and describing figures in the figure caption. AMT might edit the manuscript, but it is important to ensure that the statements are clear and unambiguous. Please check the statements that I have listed below and improve them as necessary for clarity. I look forward to receiving your revisions soon. Best Regards, Lok*
→ All the authors would like to thank you for your dedication and the remarks you suggest to improve this article. You will find specific answers hereafter to all the remarks and suggestions you made.

**2.** *Title: Shouldn't it be "meteorological biases" instead of "meteorology biases"?*
→ You were right, we have made this change in the revised manuscript.

**3.** *Line 11: Please revise the statement: "To circumvent this issue, we propose to either..... both".*
→ We proposed a new formulation in the abstract (L. 10-12); we hope it is clearer.

**4.** *Line 21: Please revise the statement: "It is found that discrepancies between two plume images due to wind direction errors in the meteorological conditions are less penalised by our new metrics with the upstream correction than without, thus avoiding the double penalty issue."*

→ Indeed, the statement is quite long and hence confusing. We have modified the sentence (L. 17-20).

**5.** *I am aware that some people use "spectra-imagery", but this can be misleading. I believe the authors meant "satellite imagery".*
→ You are right, the term "satellite imagery" is better here. We have made the changes in the revised manuscript.

**6.** *Please define the acronym "CO2M"*
→ It is now done (L. 34).

**7.** *line 40: please revise the statement: "These fast methods require only the images to provide an estimation of the emissions, but, they do so, by assuming either simplified chemistry, transport or temporal variations of the emissions."*
→ This statement is not as clear as we intended. We proposed a revised statement (L. 37-38).

**8.** *line 54: please revise the statement: "This issue is shared in other fields ....... comparisons"*
→ It should already be revised in the previous version of the manuscript. Perhaps the diff.tex we submitted did not take this specific change into account. Nevertheless, we have checked it twice and now this statement is revised.
   (L. 49-50).

**9.** *line 56: please revise the statement: "Assuming that the temporal .....inversion"*
→ This statement brings too much information at once. For the sake of clarity, we revised it (L. 50-53). We thank you for pointing this out.

**10.** *line 81: please revise the statement: "We will either consider isometry ....... compared."*
→ We made some corrections in (L. 62-65). We hope that is clearer now.

**11.** *line 104: please revise the statement: "In the present article, we focus on two-dimensional images – typically of the total column of CO2 concentration, or of ground level concentration field –, full (no mask due to filtered data or clouds), with a discretisation of N pixels".*
→ Indeed, it is insufficiently clear and yet critical. We have split the sentence in the revised manuscript (L. 86-88).

**12.** *line 144: please revise the statement: "The idea is that, instead of considering the cost of the translation, the metric adds the cost to set to zero all pixels from the first Gaussian puff to the cost to enhance the pixels at the translated location."*

→ The description of the double penalty issue has been revised (L. 126-128). We hope that it is clearer now.

**13.** *line 149: please revise the statement - it looks incomplete sentence: "More practically, the difference between the sum of the compared image pixels."*
→ We have corrected the sentence (L. 132).

**14.** *line 176: please define the acronym: "L-BFGS"*
→ It is now done (L. 157).

**15.** *line 228: please revise the statement: "This shows that the Wasserstein distance w is a softer metric ............ positions."*
→ Thank you for pointing this issue out. This statement should have been already revised in the previous manuscript. We hope this statement (L. 209-210) is clearer.

**16.** *line 254: I cannot see any equation with the division sign as mentioned here: "where ÷ is the entry-wise division in RN."*
→ Indeed, you were right, it was a writing mistake. It is now corrected.

**17.** *Page 13, Algorithm 1: I suggest to move this algorithm in the appendix section as I see it as a distraction.*
→ As you have suggested, it has been moved to the appendix.

**18.** *line 339: please define the acronym "POT".*
→ It is now done (L. 320).

**19.** *Figure 5: please expand the figure caption. It does not mention what the box, whisker, and red lines are representing. What are x- and y-axis and what are the units?*
→ Thank you for pointing this issue out. The caption is now revised. We hope it brings all the information required to understand it.

**20.** *line 386: please revise the statement and please pay attention to "between 50 km and km" in this sentence: "The horizontal grid resolution of the simulation domain (longitude: 6.82°W to 19.18°N; latitude: 42.0°N to 56.39°N, Fig. 6, Santaren et al., 2021) varies between 50km and km".*
→ Thank you for pointing out this mistake. The grid resolution goes from 50km down to 2km. The statement is now revised (L. 365).

**21.** *Figure 6: please expand the figure caption to describe all we can see in the figure. What do you mean by "Domain Santaren and al." mean?*
→ As you suggested, the caption is lacking some information. We have completed the caption.

**22.** *line 383: please revise the statement: "Consequently, the later in the day the plumes are emitted the shorter they are tracked."*
→ This part is critical since it provides a description of the plume database. We revised it (L. 376-380).

**23.** *line 394: please make sure that the reference is cited correctly. "Denier" sounds like the first name of the author.*
→ We have checked it. His full name is Hugo A. C. Denier van der Gon, Hugo being the first name and Denier van der Gon as the last name.

**24.** *line 399: please revise the statement for clarity: "We ensure that the same daily profile is applied to the source emission, then for a given hour of the day, the difference between two simulated plumes is the meteorological state".*
→ We made some modifications (L. 381-382). We hope it is clearer now.

**25.** *Figure 7: please expand the figure caption describing the box, whisker, and the colored line. Also please check what y- and x-axes are and what the units are.*
→ As it was done for Figure 5, we have expanded the caption.

**26.** *line 416: please revise the statement: "which is why we do not considered thereafter wnum in our comparisons".*
→ We have revised this statement (L. 399).

**27.** *line 418: You may have meant "as" in place of "than".*
→ Yes indeed, thank you for pointing this out.

**28.** *line 537: please revise the statement "For the case of the pixel-wise distance with an upstream correction, this can have an impact on the optimum". Optimum what?*
→ It was indeed unclear. We are talking about the optimal isometry used in $d_F$. We revised the statement (L. 519-520).

**29.** *line 545: "behaviour smoother". Did you mean "smoothing behavior"? Please revise the statement as necessary.*
→ It is obviously misleading; we meant that it penalises the position error linearly whereas the usual local metric grows fast and saturates at the end. We changed it (L. 526-527).